# Water decontamination via nonradical process by nanoconfined Fenton-like catalysts

Tongcai Liu [1], Shaoze Xiao[1], Nan Li[1], Jiabin Chen [1] ✉, Xuefei Zhou[1], Yajie Qian[2], Ching-Hua Huang[3] & Yalei Zhang [1] ✉

There is an urgent need to develop effective and sustainable solutions to reduce water pollution. Heterogeneous Fenton-like catalysts are frequently used to eliminate contaminants from water. However, the applicability of these catalysts is limited due to low availability of the reactive species (RS). Herein, nanoconfinement strategy was applied to encapsulate short-lived RS at nanoscale to boost the utilization efficiency of the RS in Fenton-like reactions. The nanoconfined catalyst was fabricated by assembling $Co_3O_4$ nanoparticles in carbon nanotube nanochannels to achieve exceptional reaction rate and excellent selectivity. Experiments collectively suggested that the degradation of contaminants was attributed to singlet oxygen ($^1O_2$). Density functional theory calculations demonstrated the nanoconfined space contributes to quantum mutation and alters the transition state to lower activation energy barriers. Simulation results revealed that the enrichment of contaminant on the catalyst reduced the migration distance and enhanced the utilization of $^1O_2$. The synergy between the shell layer and core-shell structure further improved the selectivity of $^1O_2$ towards contaminant oxidation in real waters. The nanoconfined catalyst is expected to provide a viable strategy for water pollution control.

Water pollution poses a great threat to human health and aquatic ecosystems; thus the development and implementation of wastewater treatment technologies are essential to eliminate the pollutants[1,2]. Engineered nanomaterials have been implemented in water decontamination as Fenton-like catalysts due to the excellent adsorption, catalytic, or antimicrobial properties[3–5]. To date, tremendous efforts have been devoted to design nanocatalysts and to elaborate the reaction characteristics (e.g., kinetics and mechanisms) of heterogeneous Fenton-like systems[5]. Nevertheless, the mass transfer limitation is the inherent defect of heterogeneous reactions, yet this issue has been largely overlooked in Fenton-like reactions. The ultrashort-lived radicals ($10^{-6}$–$10^{-9}$ s) hinder their mass transfer from the generation sites on the catalyst surface to the target pollutants in bulk solution, severely limiting their utilization in the heterogeneous reactions[6]. Moreover, radicals would be inevitably consumed by the water matrices (e.g., dissolved organic matter and coexisting inorganic ions), significantly reducing their reaction with target pollutants[7,8]. Consequently, there is an urgent need for nanotechnologies to enhance the efficient utilization of radicals in heterogeneous Fenton-like systems.

Encapsulating short-lived radical species and reactants within the critical diffusion length scale through spatial confinement may provide a promising strategy to overcome the challenges in heterogeneous Fenton-like reactions. Some pioneering works have reported the development of nanoconfined catalysts in the chemical synthesis, energy storage, electronics, and medical diagnostics[9–11]. The excellent performance of nanoconfined catalysts in diverse fields also brings

[1]State Key Laboratory of Pollution Control and Resources Reuse, College of Environmental Science and Engineering, Tongji University, Shanghai 200092, P. R. China. [2]College of Environmental Science and Engineering, Donghua University, Shanghai 201620, P. R. China. [3]School of Civil and Environmental Engineering, Georgia Institute of Technology, Atlanta, GA 30332, USA. ✉e-mail: chenjiabincn@163.com; zhangyalei@tongji.edu.cn

exciting opportunities for water treatment[12]. For instance, Co-TiOx nanosheets assembled within 4.6 Å 2D laminated membrane channels demonstrated remarkable reactivity towards ranitidine degradation[12]. Metal−organic framework derived yolk−shell Co/C nanoreactors exhibited selective removal of organic pollutants owing to the confinement effect[13]. $Fe_2O_3$ nanoparticles confined inside the cavity of $TiO_2$ hollow spheres showed high activity and broad pH suitability on pollutants degradation in the photo-Fenton reactions[11]. In the conventional catalytic reactions, it is generally recognized that the reactive oxygen species (ROS) generated from the active site migrate to the target pollutant at the diffusion rate[14,15]. In contrast to behavior and reaction pathways in the bulk phase, nanoconfinement alters the interaction of guest ions/molecules with reactive surfaces, leading to altered kinetics of various redox reactions and requirements for re-understanding surface/interface processes and chemical reactions. For example, Yang et al. reported the singlet oxygen ($^1O_2$)-mediated Fenton-like process under spatial nanoconfinement[16]. This finding was in stark contrast with the conventional Fenton-like processes, while the underlying reaction mechanism remains to be further revealed, specially how to regulate the reaction pathway, reaction kinetics and selectivity are still largely unclear.

Carbon nanotubes (CNTs) feature well-defined inner hollow cavities structure and unique electronic tuning properties, offering an ideal alternative for intriguing nanoconfinement environment[17]. The large surface area of hollow nanostructures provides abundant accessible surface sites with high unsaturated-coordination, facilitating the transport of small molecules in nanochannels and their concentration inside hollow structures[18]. Theoretical studies show that the concave surface is more feasible for adsorption than the convex surface, and some electron-rich reactant molecules tend to enrich on the interior surfaces[19]. Nonetheless, the pristine CNTs generally exhibit poor catalytic activity towards oxidants. Various strategies have been developed to improve the intrinsic activity of catalytic sites. Among them, the introduction of specific metal centers into CNTs can deplete/enrich the electron density of CNTs, tuning the electronic structure of the reaction sites to facilitate peroxide activation kinetics[20]. Moreover, the core-shell or core-sheath type catalysts with finite-field characteristics by inserting the metal active center into CNTs channel can improve the dispersion of active sites and inhibit the sintering of metal particles, thus ultimately obtaining excellent catalytic activity[21].

Herein, the confined growth strategy was used to precisely regulate $Co_3O_4$ nanoparticles ($Co_3O_4$ NPs) confined in CNTs (i.e., $Co_3O_4$-in-CNT). Their catalytic performance was further examined in the Fenton-like reactions, e.g., activation of hydrogen peroxide ($H_2O_2$), peroxymonosulfate (PMS), persulfate (PDS), and peracetic acid (PAA). PAA is an emerging oxidant and disinfectant for wastewater treatment due to limited harmful disinfection byproduct formation[22,23]. PAA based Fenton-like systems have received increasing attention in degradation of refractory pollutants owing to the generation of powerful organic radicals (R-O$^•$), e.g., acetoxy(per) radicals ($CH_3C(O)O^•$ and $CH_3C(O)OO^•$)[22–26]. Furthermore, the R-O$^•$ possess longer half-life and exceptional contaminants selectivity than inorganic radicals, rendering the mass transfer effortless in Fenton-like process[27]. Hence, PAA was selected as the oxidant for the in-depth exploration of the spatial nanoconfinement in Fenton-like reactions. The underlying mechanism in the confined systems was elucidated based on thermodynamic and kinetic methods from macroscopic, mesoscopic to microscopic dimensions. The mass transfer and chemical kinetic model was further developed to quantify the crucial chemical reactions, and elucidate the mass transfer process in the heterogeneous Fenton-like reactions. Finally, we investigate the stability and environmental robustness of nanomaterials in practical applications and propose the potential strategies for reaction selectivity. Overall, this work enriches the underlying mechanisms of nanoconfinement effect, paving the way for rapid and selective water purification.

## Results

### Fabrication and characterization of $Co_3O_4$-in-CNT
The preparation process of $Co_3O_4$-in-CNT and $Co_3O_4$-out-CNT were systematically expounded in Fig. 1a. The high-resolution TEM (HRTEM) clearly reveals that $Co_3O_4$ NPs were dispersed on the inner and outer surface of CNTs in $Co_3O_4$-in-CNT and $Co_3O_4$-out-CNT (Fig. 1b, d). Furthermore, the distribution and particle size of $Co_3O_4$ NPs were characterized by high-angle annular-dark field transmission electron microscopy (TEM) with energy-dispersive X-ray spectroscopy (EDX) elemental mapping. Results show that $Co_3O_4$ NPs were aligned inside the channels of CNTs in $Co_3O_4$-in-CNT (Supplementary Fig. 1c), and were randomly dispersed on the outer surface of CNTs in $Co_3O_4$-out-CNT (Supplementary Fig. 1d). Size distribution diagrams show that the $Co_3O_4$ NPs had the average sizes of 6.6 ± 0.1 nm and 8.5 ± 0.2 nm in $Co_3O_4$-in-CNT and $Co_3O_4$-in-CNT, respectively (Fig. 1f, g). X-ray diffractions (XRD) diffraction peak intensities of $Co_3O_4$ NPs in $Co_3O_4$-out-CNT and $Co_3O_4$-in-CNT were lower than those of pure $Co_3O_4$ NPs sample (Fig. 1h), indicating that $Co_3O_4$ NPs exhibited smaller crystallinity and lower average crystallite size after interaction with CNTs[28]. Moreover, the signal intensity of $Co_3O_4$ NPs in $Co_3O_4$-in-CNT slightly lower than that in $Co_3O_4$-out-CNT because $Co_3O_4$ NPs is encapsulated in CNTs shells[29]. The Brunauer−Emmett−Teller (BET) specific surface areas were determined to be 180.8 $m^2 g^{-1}$ and 164.4 $m^2 g^{-1}$ for $Co_3O_4$-in-CNT and $Co_3O_4$-out-CNT, respectively (Supplementary Fig. 2). Notably, the nitrogen adsorption−desorption isotherm of $Co_3O_4$-in-CNT presents an obvious increase at $P/P_0 > 0.8$, which can be ascribed to the presence of interior cavity created by core and shell[13].

Raman spectroscopy is always used to investigate the interaction between encapsulated metal oxide NPs and CNTs[30–32]. Two significant peaks were observed at around 1347 and 1582 $cm^{-1}$ in Raman spectra of CNTs and $Co_3O_4$-CNTs, indicating the degree of point defect ($I_D$) and graphitic structures ($I_G$), respectively (Supplementary Fig. 4). The higher value of $I_D/I_G$ for $Co_3O_4$-in-CNT indicated the formation of encapsulated $Co_3O_4$ NPs, which led to a relatively lower graphitization degree (Fig. 1i). Three new Raman bands appear at 481, 520, and 687 $cm^{-1}$ in $Co_3O_4$-out-CNT (Supplementary Fig. 5), which could be assigned to the A1g (481 $cm^{-1}$) and Eg (520 and 687 $cm^{-1}$) vibration modes of Co-O in $Co_3O_4$ NPs. Moreover, the three bands of $Co_3O_4$ NPs inside the channels of CNTs were blue shifted compared to those in $Co_3O_4$-out-CNT, implying the presence of interactions between the confined metal oxides and CNTs (Supplementary Fig. 5). Indeed, the π-electron density distribution of CNTs is not uniform due to the deviation from planarity, and the electron density of the interior surface is higher than that of the outer surface[18]. Attraction of anionic oxygen (Co-O) in $Co_3O_4$ NPs could partially compensate for the loss of electron density, resulting in the distortion of the $Co_3O_4$ lattice and the formation of non-stoichiometric oxygen vacancies. This oxygen vacancy is responsible for the shift of $v_{Co-O}$ to high frequency. Hence, the upward shift of the $v_{Co-O}$ mode was attributed to the unique interaction between the $Co_3O_4$ NPs and the inner wall of CNTs. Meanwhile, the signal intensity of $Co_3O_4$ in $Co_3O_4$-in-CNT significantly lower than that of $Co_3O_4$-out-CNT, indicating the encapsulation of $Co_3O_4$ NPs by CNTs shells.

### Fenton-like catalytic performance
The removal of bisphenol A (BPA) was negligible within 15 min by PAA alone. The degradation rate of BPA was ~50% and 52.6% in the $Co_3O_4$/PAA and $Co_3O_4$-out-CNT/PAA system, respectively (Fig. 2a). Remarkably, BPA was completely degraded in the $Co_3O_4$-in-CNT/PAA, indicating that the spatial nanoconfinement of $Co_3O_4$ sites drastically boosted the activation of PAA. Meanwhile, BPA degradation in the CNT/PAA system was much slower than that in the $Co_3O_4$-in-CNT/PAA, implying the significant role of Co sites in Fenton-like catalysis. Impressively, the degradation of BPA was greatly reduced to ~7% in the

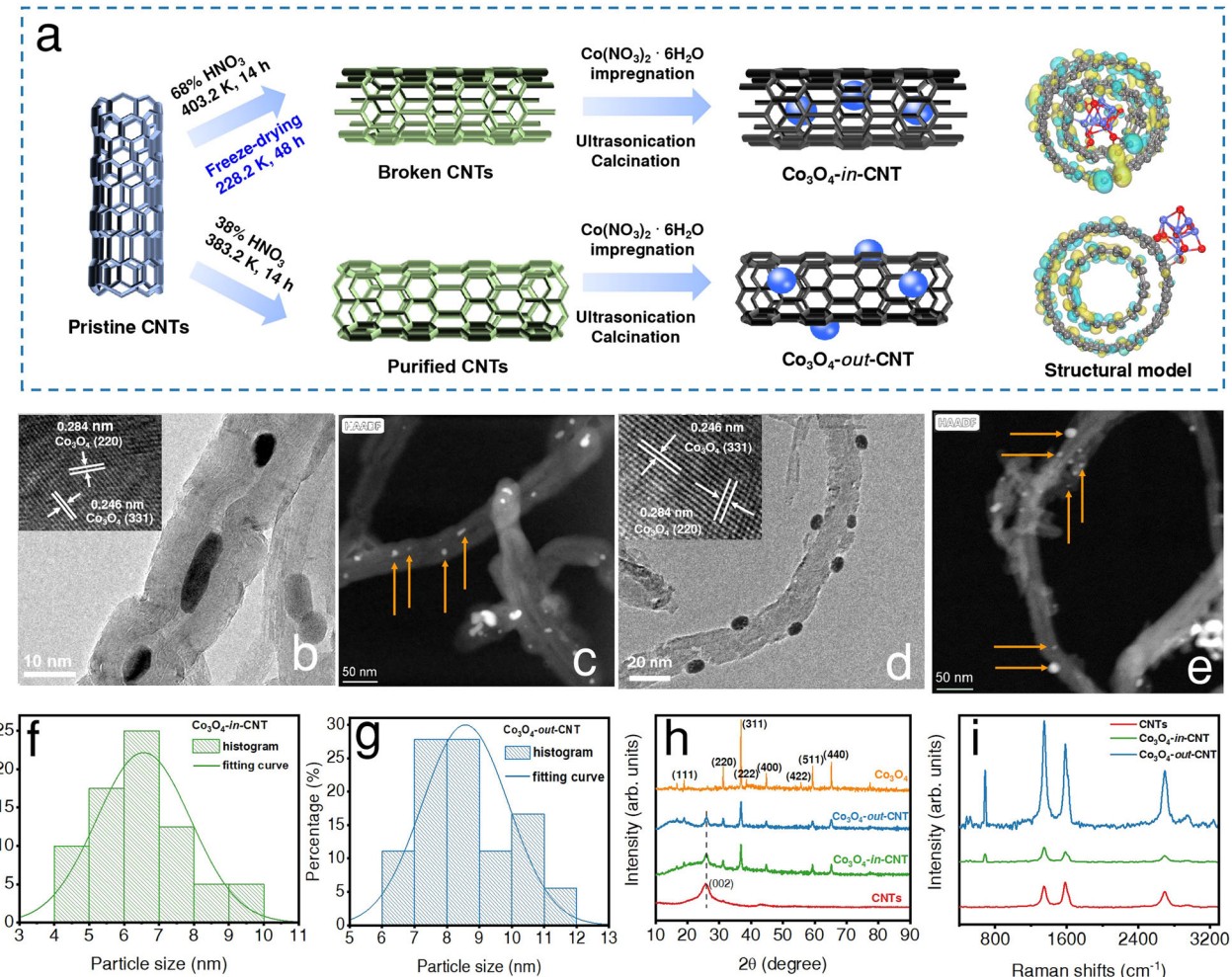

**Fig. 1 | Synthetic illustration and structural characterizations of nanocatalysts. a** Schematic illustration of the preparation strategy for $Co_3O_4$-*in*-CNT and $Co_3O_4$-*out*-CNT catalysts. **b**, **d** HRTEM images of the $Co_3O_4$-*in*-CNT and $Co_3O_4$-*out*-CNT, respectively (inset: HRTEM images of $Co_3O_4$ NPs). **c**, **e** HADDF-STEM images of $Co_3O_4$-*in*-CNT and $Co_3O_4$-*out*-CNT, respectively (the yellow arrows represent partially loaded $Co_3O_4$ NPs). **f**, **g** Size-distribution histograms of the $Co_3O_4$ NPs in $Co_3O_4$-*in*-CNT and $Co_3O_4$-*out*-CNT, respectively (the filled histogram shows the proportions of different nanoparticle diameters (the number of nanoparticles measured is about 30), and the lines are fitted normal distribution curves). **h** XRD patterns of CNTs, $Co_3O_4$, $Co_3O_4$-*out*-CNT and $Co_3O_4$-*in*-CNT. **i** Raman spectra of CNTs, $Co_3O_4$-*out*-CNT and $Co_3O_4$-*in*-CNT.

$Co_3O_4$-*out*-CNT/PAA after ten repeated cycles, but was maintained above 95% in the $Co_3O_4$-*in*-CNT/PAA system (Fig. 2b). The concentration of the leached Co ions after 150 min was ~90.2 and 52.9 μg $L^{-1}$ in the $Co_3O_4$/PAA and $Co_3O_4$-*out*-CNT/PAA, respectively. However, the leached Co was remarkably reduced to 0.5 μg $L^{-1}$ in the $Co_3O_4$-*in*-CNT/PAA system, which is far below the WHO benchmark for drinking water quality (Supplementary Fig. 7)[24,33]. Hence, the progressive deactivation of $Co_3O_4$-*out*-CNT could be primarily attributed to the considerable leaching of Co, and the competitive adsorption of BPA and its intermediates was small (Supplementary Fig. 8). The activation of other peroxides (i.e., PMS, PDS and $H_2O_2$) were further examined to verify the generic applicability, and $Co_3O_4$-*in*-CNT exhibited higher performance for activating the peroxides (Supplementary Note 5). It is worthwhile mentioning that the Fenton-like performance of $Co_3O_4$-*in*-CNT is superior to most of the state-of-the-art heterogeneous catalysts (Supplementary Table 3). Overall, the spatial confinement of $Co_3O_4$ NPs inside CNTs could remarkably reduce the leaching of Co, and also enhance Fenton-like reactivity.

### Identification of ROS in Fenton-like reactions

Previous literatures have reported that HO• and R-O• (e.g., $CH_3C(O)O$•, $CH_3C(O)OO$•, $CH_3OO$•, and $CH_3$•), can be generated from PAA

activation (Supplementary Note 6)[22–24]. The BPA degradation was not inhibited after addition of *tert*-butanol (TBA) ($k_{HO•}$ = (3.8−7.6) × $10^8 M^{-1} s^{-1}$)[34], indicating the absence of HO• in the $Co_3O_4$-*out*-CNT/PAA system (Supplementary Fig. 10). 2,4-hexadiene (2,4-HD) has been reported as an ideal quenchers for $CH_3C(O)O$• and $CH_3C(O)OO$• ($10^8–10^9 M^{-1} s^{-1}$)[35, 36]. The addition of 2,4-HD obviously retard BPA degradation (Supplementary Fig. 10), implying the generation of $CH_3C(O)O$• and $CH_3C(O)OO$•. Furthermore, electron paramagnetic resonance (EPR) analysis and contribution calculation of R-O• for BPA degradation (Supplementary Note 7,8) verified $CH_3C(O)OO$• as the primary ROS responsible for BPA degradation in the $Co_3O_4$-*out*-CNT/PAA system (Fig. 2c).

In the $Co_3O_4$-*in*-CNT/PAA system, EPR and quenching experiments (e.g., TBA or 2,4-HD) were also conducted, and results implied that the radicals, e.g., HO• and R-O•, were not responsible for BPA degradation (Supplementary Fig. 12). However, the signals of DMPO-$O_2$•‾ adducts was observed, indicating the generation of superoxide radical ($O_2$•‾) (Supplementary Fig. 13a). Nitroblue tetrazolium was regarded as a qualitative probe of $O_2$•‾ as it could be reduced by $O_2$•‾ ($k = 5.88 \times 10^4 M^{-1} s^{-1}$) to form monoformazan, exhibiting a characteristic peak at 530 nm[37]. The generation of monoformazan was observed, while the concentration was almost unaffected after

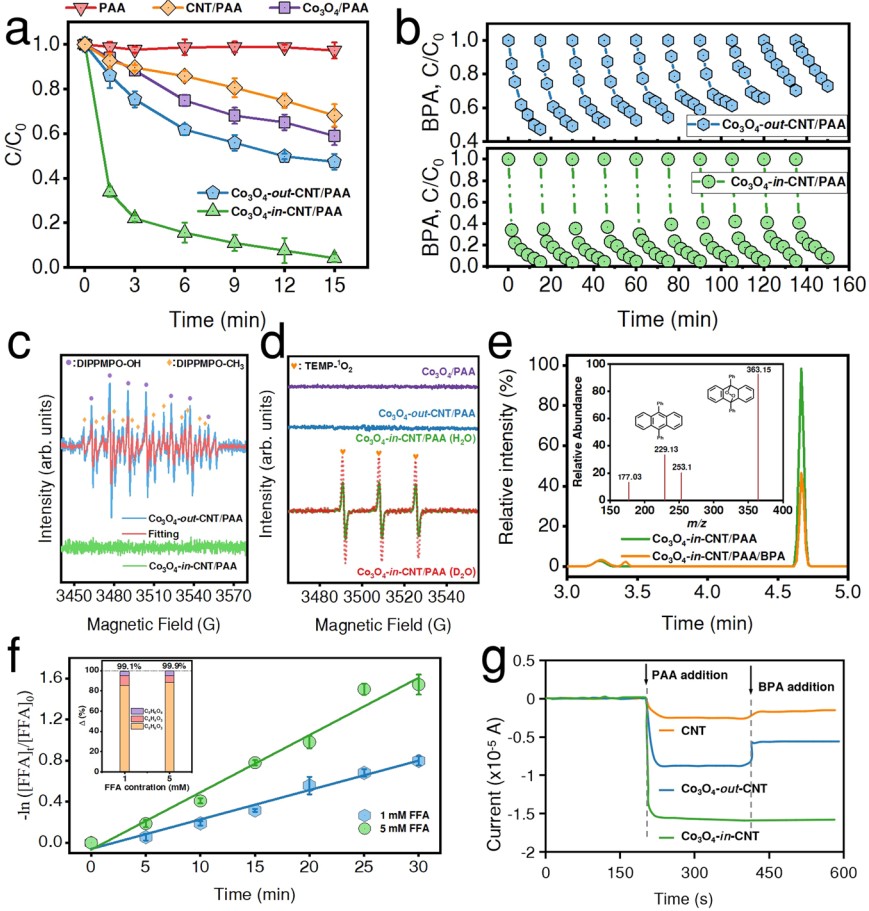

**Fig. 2 | Catalytic performance and ROS production in the Fenton-like systems.** **a** Removal of BPA in various reaction systems. **b** Cyclic performance test of nanometer catalyst. **c** Experimental and simulated EPR spectra for R-O• detection with 5-Diisopropoxyphosphoryl-5-methyl-1-pyrroline N-oxide (DIPPMPO) as trapping-agent (purple dots and yellow diamonds indicate the DIPPMPO-OH and DIPPMPO-CH₃ adducts, respectively). **d** EPR analysis for $^1O_2$ detection with 2,2,6,6-tetra-methyl-4- piperidinol (TEMP) as trapping-agent (yellow hearts indicate the TEMP-$^1O_2$ adducts). **e** Liquid chromatography-mass spectrometry of DPAO₂ from the oxidation of DPA in the Co₃O₄-*in*-CNT/PAA system (insert: the mass spectra of DPAO₂). **f** The oxidation of FFA in the Co₃O₄-*in*-CNT/PAA system (insert: the mass balance of FFA oxidation). **g** Current responses after the sequential injection of PAA and BPA at different working electrodes. The error bars in the figures represent the standard deviations from triplicate tests.

addition of BPA suggested that $O_2^{•-}$ did not directly participate in BPA degradation (Supplementary Note 10). Alternatively[1], $O_2$ always serves as crucial reactive species during the peroxide activation[38–40]. The kinetic solvent isotope effect was frequently employed to validate $^1O_2$ because the lifetime of $^1O_2$ in $H_2O$ (2.9–4.6 μs) was much shorter than that in deuterium oxide ($D_2O$) (22–70 μs)[41–43]. The results show that the degradation of BPA was promoted (Supplementary Fig. 12) and the TEMP-$^1O_2$ signal intensity significantly enhanced when the solvent was replaced by $D_2O$ (Fig. 2d). In addition, 9, 10-diphenylanthracene (DPA) shows the rapid and specific reactivity towards $^1O_2$ ($k = 1.3 \times 10^6 M^{-1} s^{-1}$), generating a stable DPA endoperoxide (DPAO₂)[16,44,45]. As shown in Fig. 2e, the signal of DPAO₂ was significantly reduced after addition of BPA, which confirmed $^1O_2$ as the reactive species for BPA degradation. Furfuryl alcohol (FFA, $C_5H_6O_2$) shows specific high reactivity towards $^1O_2$, yielding three typical products (i.e., $C_5H_6O_4$ (m/z 130), $C_4H_4O_3$ (m/z 100), and $C_5H_6O_3$ (m/z 114))[16]. Mass balance calculation indicated that FFA was converted to three classical products with a conversion rate greater than 99% (Fig. 2f), verifying the exclusive role of $^1O_2$ as the reactive species in the Co₃O₄-*in*-CNT/PAA system (Supplementary Note 11). To verify the generic applicability of the nanoconfined catalysis, the performances of multiple peroxides (i.e., $H_2O_2$, PMS, and PDS) activation were further investigated, which are discussed detail in Supplementary Note 12.

## The origin of $^1O_2$ production under spatial nanoconfinement

Dissolved oxygen (DO) and PAA were previously reported as the potential precursors for $^1O_2$ generation. The degradation of BPA was unaffected after introduction of nitrogen or oxygen gas, indicating that DO was not involved in the production of $^1O_2$ in the Co₃O₄-*in*-CNT/PAA system (Supplementary Fig. 17). Generally, $^1O_2$ can be generated from the Haber–Weiss reaction, while this pathway could be excluded because HO• was not detected in the reaction system[46,47]. Furthermore, the TEMP-$^1O_2$ signal was evidently weakened after the addition of superoxide dismutase (SOD), indicating $O_2^{•-}$ as the intermediate of $^1O_2$ production (Supplementary Fig. 18). However, considerable degradation of BPA was still achieved after scavenging of $O_2^{•-}$ (Supplementary Fig. 12), implying that there are other precursors to produce $^1O_2$ besides $O_2^{•-}$.

Recently, high-valent cobalt-oxo species [Co(IV)-OH] was reported to generate by heterolytic cleavage of the O–O bond and might serve as crucial intermediates for $^1O_2$ production[43,48,49]. To determine whether Co(IV) was produced in the Co₃O₄-*in*-CNT/PAA system, Dimethyl sulfoxide (DMSO) and methyl phenyl sulfoxide (PMSO), typical scavenger for Co(IV), were added to the reaction system. Results show that the degradation of BPA was inhibited with addition of DMSO (Supplementary Fig. 12), confirming the vital role of Co(IV) in BPA degradation. Nevertheless, PMSO₂ was gradually generated and the yield of η (PMSO₂) (the molar ratio of Δ[PMSO₂]/Δ[PMSO]) was

~36% (Supplementary Fig. 19b)[48]. Further experiments implied that Co(IV) was not directly responsible for the degradation of BPA (Supplementary Note 13). In addition, chronoamperometry measurements confirmed the absence of direct reaction between BPA and Co (IV) (Fig. 2g). Hence, $^1O_2$ was originated from the transformation of $O_2^{\cdot-}$ and Co(IV) in the $Co_3O_4$-*in*-CNT/PAA system.

## Activation process under spatial nanoconfinement

The activation of PAA was previously reported to proceed through the redox cycle of Co(II)/Co(III) to produce R-O[22,23], which resembled the $Co_3O_4$-*out*-CNT/PAA process (Supplementary Note 6). In the $Co_3O_4$-*in*-CNT/PAA system, PAA activation proceeded via a non-radical process (i.e., $^1O_2$) rather than the radical process. First, Co(II) is coordinated with PAA to form $Co(II)$-$CH_3C(O)OOH$ complex, which is then decomposed to produce Co(IV) through two-electron transfer (Eq. (1) and Eq. (2))[43]. Afterwards, Co(IV) reacts with PAA to generate $^1O_2$ (Eq. (3)), and the coexisting $H_2O_2$ could also accelerate the consumption of Co(IV) and production of $O_2^{\cdot-}$ (Eq. (4))[50]. Moreover, the generated $O_2^{\cdot-}$ further participated in the production of $^1O_2$ through multiple pathways. On the one hand, the recombination of $O_2^{\cdot-}$/$HO_2^{\cdot}$ could produce $^1O_2$, and this process is thermodynamically spontaneous based on its Gibbs free energy of $-6.4$ kcal mol$^{-1}$ (Eq. (6))[18,51]. On the other hand, $O_2^{\cdot-}$ can be oxidized by high valent Co species (i.e., Co(IV)/Co(III)) to generate $^1O_2$, and concomitant reduction of Co(III) to Co(II), respectively (Eq. (5))[16,18,52]. Intriguingly, this reaction mechanism is in stark contrast to the traditional Fenton-like processes. This Fenton-like reaction mechanism under spatial nanoconfinement was elucidated from the perspectives of thermodynamics and kinetics in the following sections (Fig. 3a).

$$\equiv Co(II) + CH_3C(O)OOH \rightarrow \equiv Co(II) - CH_3C(O)OOH \text{ (complex)} \quad (1)$$

$$\equiv Co(II) - CH_3C(O)OOH \text{ (complex)} \rightarrow \equiv Co(IV) + CH_3C(O)O^- + OH^- \quad (2)$$

$$2CH_3C(O)OOH + \equiv Co(IV) + H_2O \rightarrow 2CH_3C(O)O^- + Co(III) + 3H^+ + {}^1O_2 \quad (3)$$

$$H_2O_2 + \equiv Co(IV) \rightarrow O_2^{\bullet-} + Co(III) + 2H^+ \quad (4)$$

$$O_2^- + \equiv Co(IV)/Co(III) \rightarrow \equiv Co(III)/Co(II) + {}^1O_2 \quad (5)$$

$$2O_2^- + 2H^+ \rightarrow H_2O_2 + {}^1O_2 \quad (6)$$

## Theoretical study on the regulation mechanism

The rationality for regulation mechanism was further inspected by density functional theory (DFT) calculations. Adsorption energies ($E_{ads}$) calculation indicated that the confinement of $Co_3O_4$ NPs inside CNTs strongly enhanced the adsorption capacity of PAA (Supplementary Fig. 21). Meanwhile, the length of O–O and C–O bonds of PAA adsorbed on the catalysts increased, further providing the evidence that $Co_3O_4$-*in*-CNT promoted PAA activation (Supplementary Table 6). The electrostatic potentials mapping was used to reflect the charge distribution of CNTs before and after $Co_3O_4$ NPs anchoring (Fig. 3c)[53]. After anchoring Co3O4 NPs inside CNTs, the charge distribution and surface chemistry became uneven, and the $Co_3O_4$ site had positive potentials that could effectively activate PAA to produce ROS. In addition, the lower energy barrier of $Co_3O_4$-*in*-CNT (0.071 eV) was observed, implying that electron migration from the lower unoccupied

molecular orbital to the highest occupied molecular orbital in $Co_3O_4$-*in*-CNT is more favorable[54].

DFT calculations were performed to calculate the corresponding free energies of intermediates (INT) and energy of transition states (TS), and the thermodynamic barrier of the R-O$^{\cdot}$ or $^1O_2$ formation (Supplementary Figs. 23, 24). As illustrated in Fig. 3d, in the $Co_3O_4$-*in*-CNT/PAA system, $\equiv Co(II)$ interacted with PAA to yield $\equiv Co(II)$-$CH_3C(O)OOH$ (INT_1), in which $\equiv Co(II)$ was oxidized to $\equiv Co(IV)$. Afterwards, the $\equiv Co(IV) + CH_3C(O)O^-$ in INT_2 was formed via TS_3 with an energy barrier of 10.71 kcal mol$^{-1}$. $\equiv Co(IV)$ acted as a crucial intermediate to generate $^1O_2$ and participating in the transformation of $O_2^{\cdot-}$, and the total energy barrier was calculated to be $-10.47$ and $-28.45$ kcal mol$^{-1}$, respectively (Fig. 3d). The energy barrier of $^1O_2$ formation is lower than that of R-O$^{\cdot}$ production ($-8.10$ kcal mol$^{-1}$). In the $Co_3O_4$-*out*-CNT/PAA system, the energy profile showed that the decomposition of $\equiv Co(II)$-$CH_3C(O)OOH$ (INT_1) to generate $\equiv Co(III) + CH_3C(O)O^{\cdot}$ (INT_2) with an energy barrier of 6.58 kcal mol$^{-1}$. Then, $\equiv Co(III)$ can activate $CH_3C(O)OOH$ to produce $\equiv Co(II) + CH_3C(O)OO^{\cdot}$, with an energy barrier of $-8.20$ kcal mol$^{-1}$. Whereas, the total energy barriers were calculated to be $-\sim5.28$ and $-4.33$ kcal mol$^{-1}$ for $^1O_2$ generation in the $Co_3O_4$-*out*-CNT/PAA system (Fig. 3d). Overall, the generation of $^1O_2$ was thermodynamically favorable under spatial nanoconfinement.

## Mechanistic insight into kinetics enhancement under spatial nanoconfinement

*Reactant enrichment effect.* The reactant enrichment has been deemed as a critical manifestation of nanoconfinement effect in the microenvironment of nanoreactors. For the hollow tubular nanostructures, the void environment can offer a local space with the concave interface to accumulate reactant molecules. When the active sites are interiorly loaded, the reinforcement of local reactant concentration will deliver higher reaction rates[55]. Therefore, the contribution of the reactant enrichment effect was emphatically investigated under spatial nanoconfinement. Results manifest that the BPA adsorption efficiency was positively correlated to the Fenton-like reaction performance (Supplementary Fig. 25). Furthermore, the adsorption of BPA under acidic conditions was significantly higher than that in alkaline and neutral conditions (Supplementary Note 15), because $Co_3O_4$-*in*-CNT (pH$_{pzc}$ 3.6) with negatively charged surface preferentially adsorb the cationic BPA (pKa 10.3) via electrostatic interaction. Although the adsorption of BPA was suppressed with increasing pH, BPA still had high removal rate under alkaline conditions (Supplementary Fig. 25), implying other vital factors responsible for kinetic enhancing under spatial constraints.

*Electronic metal-support interaction effect.* The stress variation of the curved carbon shell endows a deviation of graphene layers from planarity, which induce a shift of π-electron density from the internal surface (concave) to the external surface (convex). As a result, the electron-deficient and electron-rich states are formed at interior and exterior positions of the CNTs, respectively[56]. The internally loaded $Co_3O_4$ NPs ($Co_3O_4$-*in*-CNT) contribute more electrons to the electron-deficient carbon surface (internal surface) than externally loaded $Co_3O_4$ NPs ($Co_3O_4$-*out*-CNT)[57]. Consequently, the interiorly loaded $Co_3O_4$ NPs always exhibit relatively lower electron density (electron-deficient) due to this electronic metal support interaction. Then, reactants with electron donating group (e.g., BPA and PAA) theoretically inclined to adsorb to the interiorly loaded $Co_3O_4$ NPs with relative lower electron density. In terms of intrinsic material properties, $Co_3O_4$-*in*-CNT had a substantially smaller EIS Nyquist arc diameter than that of CNTs and $Co_3O_4$-*out*-CNT. The electrochemical performance suggested that the resistance was significantly reduced after anchoring $Co_3O_4$ sites into the CNTs, elucidating the electronic metal-support interaction effect (Fig. 4a). Moreover, chronoamperometry measurements showed that $Co_3O_4$-*in*-CNT exhibited the maximum current hopping intensity after introduction of PAA (Fig. 3g), further demonstrating the efficient internal electron migration between $Co_3O_4$-*in*-CNT and PAA[27]. Hence, the

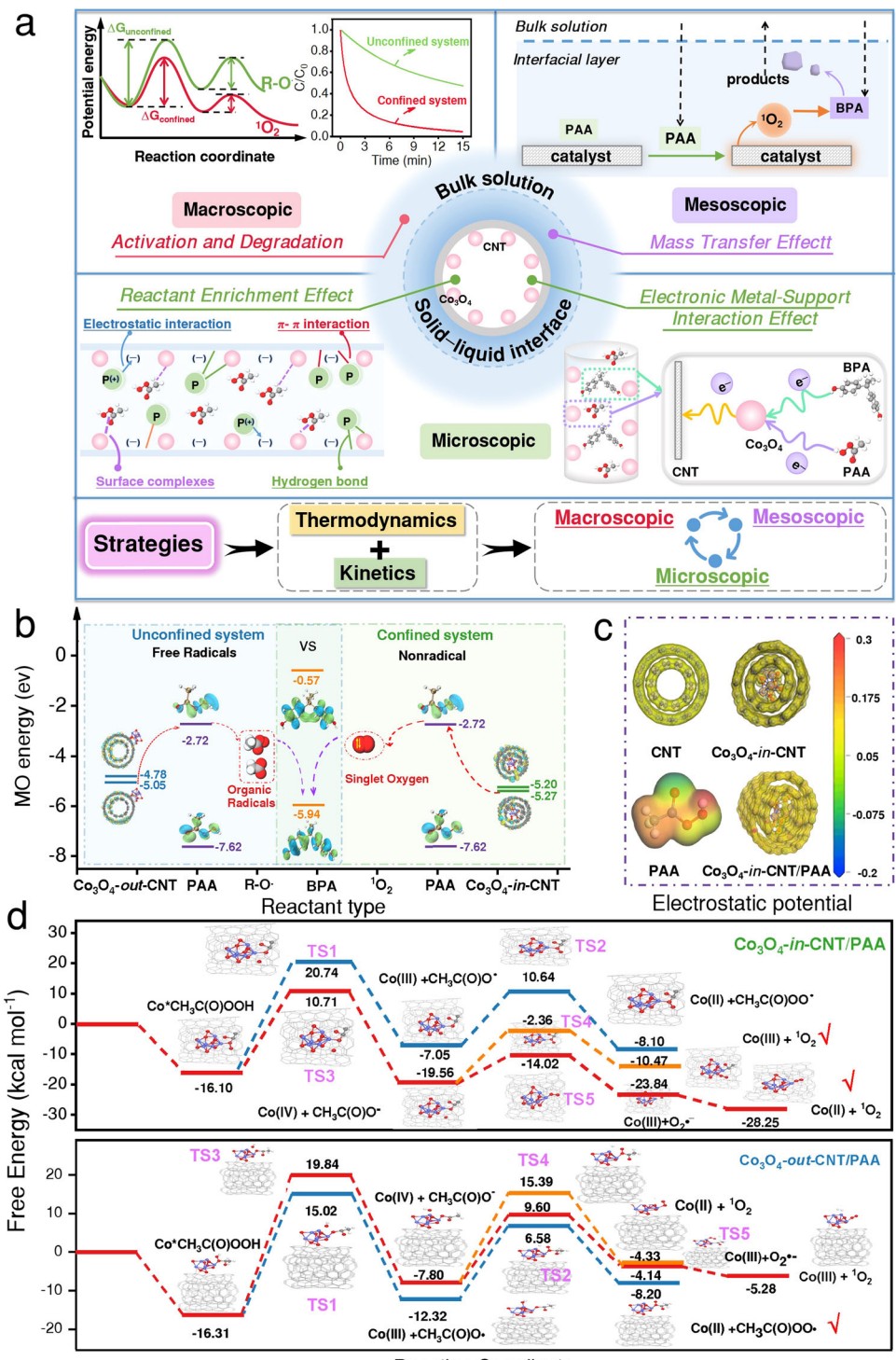

**Fig. 3 | DFT calculation unravel the mechanism of Fenton-like reaction under spatial nanoconfinement.** **a** Overview of strategies to elucidate the intrinsic principles of Fenton-like reactions in confined systems from macroscopic, mesoscopic to microscopic dimensions. **b** The proposed overall Fenton-like reaction mechanism in unconfined and confined systems. **c** Distributions of electrostatic potential (ESP) for CNTs, Co$_3$O$_4$-*in*-CNT, PAA, and interaction between Co$_3$O$_4$-*in*-CNT and PAA. **d** The free energy profiles of the proposed intermediates and TSs in the Co$_3$O$_4$-*in*-CNT/PAA and Co$_3$O$_4$-*out*-CNT/PAA system.

electronic metal-support interactions enhanced the electron migration kinetics and facilitated the production of ROS[28].

*Mass transfer effect.* Although the heterogeneous Fenton process has been widely applied in water decontamination, the mass transfer process is still poorly understood. In the heterogeneous catalysis, the produced reactive species can be localized to the catalyst surface (e.g., activated peroxide complexes, and surface-localized radicals), or

diffused from the surface in a rather limited distance[8]. Considering the ultrashort lifetime and the limited diffusion extent of ROS from the catalyst surface, the oxidation of target compounds potentially occurs in close proximity to the surface rather than in bulk solution[14]. Consequently, a heterogeneous catalytic model including fluid dynamics, mass transfer and chemical reaction kinetics was developed to better understand the spatial distribution of ROS and the solid−liquid

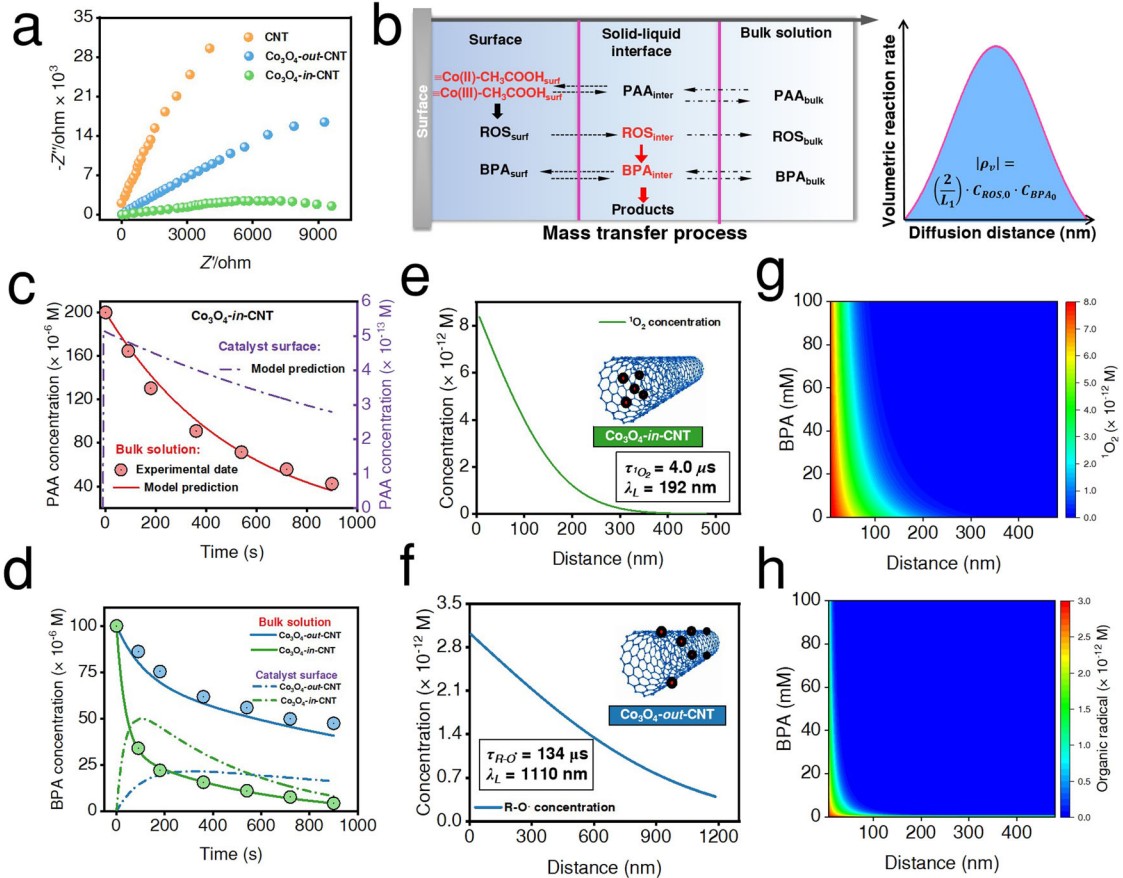

**Fig. 4 | Potential mechanism of kinetics enhancement under spatial nanoconfinement. a** EIS (Electrochemical impedance spectroscopy) Nyquist plots of CNTs, Co$_3$O$_4$-*out*-CNT and Co$_3$O$_4$-*in*-CNT. **b** The mass transfer process of species in heterogeneous Fenton system, and the relationship between mass transfer distance with reaction rate. **c, d** Variation of PAA, BPA concentrations in bulk solution and on catalyst surface, respectively (circles represent data points, solid lines represent the simulated change of species concentration in bulk solution, and dashed lines represent the simulated change of species concentration on catalyst surface). **e, f** Relationship between ROS concentration and migration-diffusion distance in the Co$_3$O$_4$-*in*-CNT/PAA and Co$_3$O$_4$-*out*-CNT/PAA system, respectively. **g, h** Two-dimensional illustration of ROS exposure as a function of BPA concentration and distance from the catalyst surface.

interface mass transfer process (Fig. 4b). It is assumed that all reactions take place in the solid−liquid interface, the concentration of ROS and BPA vary with the distance from the catalyst surface, and the distribution of ROS and BPA are as follows:

$$C_{ROS} = f(x) \tag{7}$$

$$C_{BPA} = g(x) \tag{8}$$

The concentration of ROS decreased from the catalyst surface to the bulk solution, whereas BPA concentration reduced from the bulk solution to the catalyst surface. Hence, the degradation rate of BPA in the solid−liquid interface can be expressed in integral form as:

$$\int_0^{L_1} \rho = \int_0^{L_1} k \cdot f(x) \cdot g(x) \cdot d_x \tag{9}$$

where $k$ is the second-order rate constant of the reaction between ROS and BPA, $L_1$ is thickness of the solid−liquid interface. After assuming that the concentration changes of ROS and BPA in the solid−liquid interface are approximately linear, the degradation of BPA can be expressed as follows:

$$f(x) = -k_1 x + C_{ROS,0} \tag{10}$$

$$g(x) = k_2 x \tag{11}$$

$$\left| \int_0^{L_1} \rho \right| = \left| \int_0^{L_1} k \cdot (-k_1 k_2 x^2 + C_{ROS,0} \cdot k_2 \cdot x) \cdot d_x \right| \tag{12}$$
$$= \left( \frac{2 \cdot k}{L_1} \right) \cdot C_{ROS,0} \cdot C_{BPA_0}$$

Furthermore, the diffusion process of ROS was described by the modified Fick's second law (Supplementary Note 16)[14]. The results showed that the BPA and PAA concentrations increased dramatically upon approaching to the surface due to the synergy of electronic metal-support interaction and reactant enrichment effect (Fig. 4c, d). In addition, the $^1O_2$ concentration was increased in solid−liquid interface and estimated to be ~1.5 × 10$^{-12}$ M (Supplementary Table 10). As shown in Fig. 4e, f, the ROS concentration drops sharply with increasing migration distance, due to the ultrashort lifetime of ROS and pollutant consumption. Furthermore, kinetic simulation of ROS indicated that the lifetime of $^1O_2$ ($\tau^1O_2$) and R-O$^·$ ($\tau_{R-O·}$) were predicted to be 4 μs and 134 μs in the Co$_3$O$_4$-*in*-CNT and Co$_3$O$_4$-*out*-CNT, respectively (Supplementary Note 18). Moreover, the diffusion distance of $^1O_2$ and R-O$^·$ ($\lambda_L$) were calculated to be 192 nm and 1110 nm ($\lambda_L = 2\sqrt{D_{RS} \times \tau_{RS}}$, where $D_{RS}$ is the diffusion coefficient, i.e., 2.3 × 10$^{-9}$ m$^2$ s$^{-1}$)[15], respectively. As shown in Fig. 4e, the consumption of 99% $^1O_2$ occurs within 350 nm from the surface, which was ascribed to

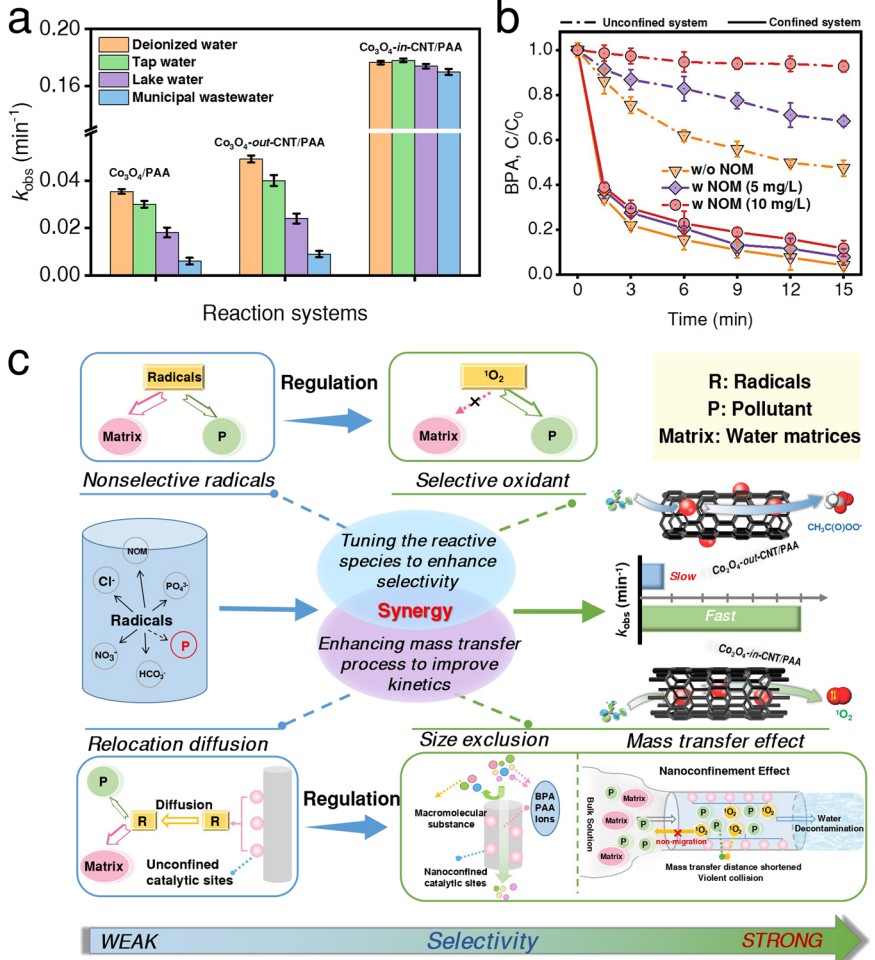

**Fig. 5 | Potential application of Fenton-like reactions, and strategies to enhance or alter selectivity. a** Influence of NOM on BPA removal in the Fenton-like systems. **b** The degradation of BPA in the real water matrices. **c** Overview of the selective oxidation strategies for Fenton-like reaction. The error bars in the figures represent the standard deviations from triplicate tests.

the increase of the interphase mass transfer resistance. Conversely, the enhanced kinetics is attributed to the nanoconfinement effect; i.e.,$^1O_2$ exposure is increased by confining short-lived species in a nanoscale domain (e.g., red-colored region in Fig. 4g with higher ROS concentration). Therefore, the confined space permits full accessibility of PAA and confines high instantaneous concentration of $^1O_2$ and BPA in a local microenvironment, which provides a driving force to facilitate the oxidation of BPA.

## Potential environmental applications

The degradation pathway and toxicity of BPA were investigated in the $Co_3O_4$-*in*-CNT/PAA system (Supplementary Note 20). The condensed Fukui functions were calculated to analyze the reaction sites of BPA, and the transformation products (TPs) and degradation pathways were proposed in Supplementary Fig. 37. The Ecological Structure–Activity Relationship (ECOSAR) class program (ECOSAR v2.2, EPA) was utilized to predict the ecotoxicity of BPA and TPs. The results suggest that the toxicity of TPs was significantly lower than BPA, which was primarily caused by the cleavage of aromatic ring (i.e., one of the key biotoxic groups) (Supplementary Table 13). Moreover, the acute toxicity of the solution after treatment was measured by the inhibition rate of luminescence against *Vibrio fischeri* bacteria. The luminescence inhibition obviously reduced to 1.5% when the reaction time was further increased to 60 min (Supplementary Fig. 38). Hence, $^1O_2$ not only effectively removed water pollutants but also selectively destroyed the biological toxicity of pollutants.

The stability of catalyst is critical factor in practical application. $Co_3O_4$-*in*-CNT exhibits excellent performance throughout each cycle and the leached Co was 0.5 μg $L^{-1}$ after reaction for 150 min, indicating the good reusability. Degradation experiment was further performed in real waters (e.g., tap water, lake water and municipal wastewater) to evaluate the potential environmental application. The excellent degradation efficiency could still achieve, highlighting the practicability of this confined Fenton-like process in water treatment (Fig. 5a). Overall, the superior performance could be ascribed to the synergetic effects from shell layer (size-exclusion) and core/shell (confinement effect), which further enhanced the selectivity of $^1O_2$ towards contaminant oxidation in real waters (Supplementary Note 21). Specifically, the majority of macromolecular compounds and colloidal water components (e.g., natural organic matter and soluble microbial products) could be excluded by size exclusion from $Co_3O_4$-*in*-CNT (Fig. 5b)[58,59]. On the other hand, the selective oxidant $^1O_2$ is less affected by the water matrices than the radicals (Supplementary Fig. 39a). To further verify the selectivity of the $Co_3O_4$-*in*-CNT catalysis, the removal performances of multiple recalcitrant pollutants of environmental and public health concerns, including pharmaceutical (sulfamethoxazole, SMX)), environmental hormone (17α-ethinylestradiol, EE2), dye (Rhodamine B, RhB), pesticide (imidacloprid, IMD) and phenols (4-chlorophenol, 4-CP) were further examined (Fig. 5d). The results confirmed that these pollutants could be efficiently removed under spatial nanoconfinement, while benzoic acid (BA), nitrobenzene (NB), carbamazepine (CBZ), and atrazine (ATZ) with electron-withdrawing groups

were resistant to degradation in the $Co_3O_4$-*in*-CNT/PAA system, suggesting that $^1O_2$ exhibits a strong selectivity towards pollutants (Supplementary Fig. 41). Overall, the excellent stability and selectivity of the nanoconfined catalysts were achieved (Fig. 5c) and then the selective oxidation strategy was proposed (Supplementary Note 21), which expanded the application of nanoconfined catalyst for water treatment.

## Discussion

In summary, we have developed an efficient nanoconfined catalyst for purifying wastewater, which demonstrate unprecedented reaction rates and selectivity in water decontamination. Electron transfer from the active metal core to the carbon layer stimulates unique catalytic activity on the carbon surface, regulating the mechanism from a classical radical process to a nonradical process in Fenton-like reactions. The intrinsic structure-activity relationship between reactive species and pollutant removal was thoroughly understood by combining experimental analyses with theoretical calculations. The kinetics enhancement is primarily attributable to the reactant enrichment, electronic metal-support interaction and mass transfer effect within spatial nanoconfinement structure. The nano-confined Fenton-like reaction presents a clear size exclusion and notable selectivity towards target pollutants, rendering the nanocatalyst a promising candidate for contaminant elimination in real water matrices. Moreover, the stable carbon layer protects the inner metal core from the destructive reaction environment, thus protecting the inner Co species from metal leaching and facilitating the mass transfer. Overall, this work provides a deep insight into the creation and effective utilization of spatial nanoconfinement in Fenton-like nanocatalysts for water purification.

## Methods

### Chemicals

Multiwalled carbon nanotubes (CNTs) (<d> = 10−20 nm and <l> = 10−30 μm) were purchased from XFNANO Nano Co., Ltd. (Nanjjing, China). Other reagents and materials are provided in the Supplementary Note 1.

### Fabrication of catalysts

The synthesis of nanocatalysts mainly includes the modification of pristine CNTs and the loading of $Co_3O_4$ NPs. First, the mixture of pristine CNTs (5 g) and 200 mL of concentrated $HNO_3$ (68 wt%) was refluxed at $403.2 \pm 5$ K for 14 h in an oil bath. Then, mixture was washed with deionized water to remove acid residues and dried in oven at 333.2 K for 12 h, followed by lyophilization (SCIENTZ-12N, China) at 228.2 K for 48 h. Afterwards, 20 mL mixed solution containing $Co(NO_3)_2 \cdot 6H_2O$ (0.3 g, 51.54 mM) and EtOH (99 wt%) was dispersed by sonication to serve as an cobalt precursor. The dispersion solution prepared above was sonicated in acetone (250 mL) solution for 4 h, which was introduced into the CNTs channels utilizing the capillary forces of CNTs aided by ultrasonic and stirring treatment. The as-obtained solid was gradually heated to 623.2 K (heating rate: 274.2 K $min^{-1}$) in air and maintained for 2 h. Finally, the $Co_3O_4$-*out*-CNT were fabricated by washed with EtOH and deionized water, and dried in a vacuum desiccator overnight. For comparison, $Co_3O_4$ NPs deposited on the outer surface of CNTs ($Co_3O_4$-*out*-CNT) was also fabricated, and the details of the synthesis process are provided in Supplementary Note 2.

### Batch experiments

The batch experiments were conducted in a 100 mL conical flask under magnetic stirring at $293.2 \pm 5$ K. Unless otherwise specified, the concentrations of BPA, PAA and catalysts are 100 μM, 200 μM, 0.1 g $L^{-1}$, respectively. Before the degradation, the catalysts were added into the BPA solution and stirred for 10 min to reach the adsorption

−desorption equilibrium. Subsequently, reactions were initiated by addition of PAA to the solution containing the catalyst and BPA. The initial pH was adjusted by diluted $H_2SO_4$ or NaOH to the designated value. Reaction aliquots were periodically collected and immediately quenched with excess sodium thiosulfate ($Na_2S_2O_3$), then filtered through a membrane (0.22 μm) to remove the solid catalysts and stored in amber vials at 5 °C before analysis. All experiments were conducted in triplicate, and the error bars in the figures represent the standard deviations from triplicate tests.

### Characterization methods

The crystal phases and structures of these samples were characterized by XRD (D8 Advance diffractometer, Bruker, Germany) with Cu Kα radiation ($\lambda = 1.5406$ Å). The composition and element chemical state of elements were determined by X-ray photoelectron spectroscopy (XPS, ESCALAB 250Xi, ThermoFisher Scientific, USA). The specified surface areas were measured using BET method with a Builder 4200 instrument (Tristar II 3020 M, Micromeritics Co., USA). Representative high-angle dark-field scanning transmission electron microscopy (HADDF-STEM) images with energy-dispersive EDX elemental mapping was conducted to characterize the dispersion and configuration of the catalysts, as well as to map the abundance of C, O, and Co in the catalyst. The insets of HRTEM were carried out via a JEM 2100F field emission transmission electron microscope at an accelerating voltage of 200 kV to characterize the morphological and textural properties. Raman spectra were recorded with a LabRAM Aramis Raman spectrometer (Horiba Scientific) with an Ar laser at 532 nm. The powdered sample was pressed into the sample holder and then placed into a home-built high-temperature reaction cell for in situ Raman measurements.

### Analytical methods

Organic contaminants were determined on a high-performance liquid chromatography (HPLC) system (1200, Agilent Technology, USA) equipped with a C18 column ($4.6 \times 250$ mm, 5 μm). TPs of BPA were analyzed on an HPLC system (1290uplc, Agilent, USA) connected with a triple quadrupole mass spectrometer (QTOF6550, Agilent, USA), the details are provided in Supplementary Note 3. EPR spectra were obtained using MS-5000 spectrometer (Bruker, Germany) (Supplementary Note 7). Kinetic solvent isotope effect, FFA and DPA product detection was used to determine $^1O_2$ production (Supplementary Note 8,9). Electrochemical characterization was performed using a CHI 760E electrochemical workstation equipped with a standard three-electrode system (Supplementary Note 14). The optimized geometry of BPA was optimized using the Gaussian 09 with a basis set of B3LYP/6−31G (d, p) and was visualized using Multiwfn[60]. The aquatic toxicity of the transformation byproducts was predicted using the USEPA ECOSAR program[26].

### Quantum chemical calculations

All of spin-polarized calculations based on DFT were performed by utilizing DMol3 package[61]. The generalized gradient approximation in the Perdew−Burke−Ernzerhof form and Semicore Pseudopotential method (DSPP) with the double numerical basis sets plus the polarization functional (DNP) were adopted[62,63]. A DFT-D correction with Grimme scheme was used to calculate the dispersion interaction[64]. The SCF convergence for each electronic energy was set as $1.0 \times 10^{-5}$ Ha, and the geometry optimization convergence criteria were set as follows: $1.0 \times 10^{-5}$ Ha for energy, 0.004 Ha $Å^{-1}$ for force, and 0.01 Å for displacement, respectively. Energy barriers were examined by linear and quadratic synchronous transit methods in combination with the conjugated gradient refinement. Spin polarizations are also considered in all calculations. The free energies ($G$) of different intermediates are defined as $G = E_i - E_{reactant}$ ($E_i$ the energy of intermediates and $E_{reactant}$ the total energy of reactants) and finally obtained by

$G = E_{total} + E_{ZPE} - TS$, where $E_{total}$, $E_{ZPE}$, and TS are the ground-state energy, zero-point energies, and entropy terms, respectively, with the latter two taking vibration frequencies from DFT.

## Kinetic modeling

The model describing various kinetic reactions involved in ROS production and BPA degradation, and coupled the mass transfer processes of related chemical species. Specifically, the mass transfer process of reactant species between catalyst surface and bulk solution was simulated by the two-film theory. In addition, the diffusion of ROS and BPA was implemented by the modified Fick's second law, and the details are provided in Supplementary Note 17. Kinetic models were developed based on the first principles using Matlab R2020a. Principal component analysis of the kinetic models were used to generate normalized sensitivity coefficients to determine a set of important reactions in the system. The optimization of fitting parameters was conducted using a custom blend of gradient descent and Levenberg–Marquardt optimization after first finding candidate solutions by Monte Carlo sampling.

## Statistics and reproducibility

All the statistical analysis was performed using Microsoft Excel (Microsoft Inc.), OriginPro 2021 (OriginLab Co, Northampton, MA) and Matlab R2020a (MathWorks, Inc.) software. All quantitative results were presented as mean ± standard deviation. All the experiments in the main text and supplementary information have been repeated three times or more independently with similar results. To show the imaging performance, representative imaging results were included in the manuscript.

## Data availability

The authors declare that all data supporting the findings of this study are available within the article and the supplementary information. Any additional data are available from the corresponding author.

## Code availability

All the data analysis codes related to this study are available from the corresponding authors.

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

## Acknowledgements
This study was supported by National Natural Science Foundation of China (52122006), Shanghai Rising-Star Program (20QC1401200), Shanghai Science and Technology Innovation Action Plan (No. 22dz1205700), and the Fundamental Research Funds for the Central Universities.

## Author contributions
T.L., J.C., and S.X. designed the experiments. T.L. and N.L. performed the experiments. J.C. and Y.Z. supervised the study and experiments. T.L. and S.X. performed the data analyses and prepared this manuscript. S.X. and Y.Q. developed and analyzed the reaction model. J.C., C.H., X.Z., and Y.Z. provided constructive suggestions for the manuscript revision.

## Competing interests
The authors declare no competing interests.
