## [Peer Review File · Nature Communications]

Reviewer comments, first round

Reviewer #1 (Remarks to the Author):

This manuscript reported a very interesting and important work on the development of advanced oxidation processes for organic pollutant degradation, which uses the nanoconfinement effect to encapsulate pollutants and reactive species within the critical diffusion length to reduce the mass transfer limitation and improve the utilization rate of reactive species. Since there are numerous studies reporting development of heterogeneous catalysts to promote Fenton-like reactions, this work takes a novel perspective on key scientific questions. Intriguingly, the effective regulation of the reaction mechanism of heterogeneous catalytic oxidation system was realized under spatial nanoconfinement, which provides a brand-new strategy for the transformation of reaction pathway from radical to nonradical in the heterogeneous Fenton-like reaction. In addition, the authors have provided a lot of fresh knowledge on the nanoconfinement effect and mass transfer models. Nonradical dominated oxidation systems show strong selectivity, and the nano-confined catalyst presents shell layer (size exclusion) and core/shell (confinement effect) further enhance the selectivity of HO_2 towards water purification, which greatly solves the practical application issues in water treatment. The manuscript is well written and the new mechanisms and phenomena are well verified by a complete and logical experiments. Both in terms of method development and mechanism interpretation, this work has obtained meaningful results in water purification. Consequently, I am sure that this work deserves publication in Nature Communications after minor revision according to the following comments:

1. The authors make an exciting statement on the mechanism of peracetic acid (PAA)-based pollutants removal systems, yet the experimental results were significantly different from current reports in the PAA-based oxidation system (a process dominated by free radicals). Considerable efforts have been made to explore the mechanism of the formation of active species in the manuscript. I'm curious about the advantages that nonradicals have over radical oxidation systems in water treatment.
2. It is very difficult to quantify the mass transfer process and the reactive species in the nanochannel through experiments. The author solves this problem through the model, which I think is a very important method. To validate this model a large amount of experimental data must be contemporarily fitted to obtain robust data and to have an idea of the sensitivity of the parameters on the recorded experimental profiles. A carefully evaluation of these points must be carried out, without this critical evaluation the conclusions obtained from the kinetic model are very few supported.
3. The formation of HO_2 is generally accompanied by the generation of free radicals in Fenton-like systems. In this work, what's the proportion of HO_2 as an active species in pollutant degradation.
4. Lines 554-573: The quenching of radicals in Fenton-like reactions by various inorganic anions or organic compounds limits their value in industrial applications, thus selective oxidation strategy is an important index in practical water decontamination. I think more discussion is needed to improve the selectivity method.
5. Some formatting and grammar comments: Line 37: "mass transfer-chemical kinetic model" should be "mass transfer and chemical kinetic model". Lines 290-291: this sentence needs to be revised. The expression forms of free radicals need to be unified in the whole manuscript.

Reviewer #2 (Remarks to the Author):

The manuscript NCOMMS-22-45381 deals with a very interesting issue, namely the enhanced removal of pollutants in confined medium. The paper is well written, it is easy to follow, results are interesting and supported by data. Findings are new and relevant.

However, I have some comments to do on the work that authors should consider:

1. I am not sure if the word Fenton (or even Fenton-like) describes the process. Fenton deals with decomposition of hydrogen peroxide into highly reactive species by iron salts in a catalytic cycle

involving Fe(II) and Fe(III), and where hydroxyl radicals play an important role. Here we have a relevant process, but involving cobalt, peracetic acid (PAA) and seems to be based on singlet oxygen. Although it is related with Fenton chemistry, I suggest changing Fenton-like by cobalt-PAA system, at least in the title

2. Authors attribute a relevant role to singlet oxygen and this result seems supported. However, this is a weakness of the system, as singlet oxygen is a worse oxidising agent than hydroxyl radicals, and a wide range of chemical moieties might be reluctant to its action. Hence, I think that it can only be useful for some target molecules.

3. The environmental relevance section should be completely rewritten. In this work only primary removal of pollutants is studied via chromatography. That means that disappearance of the pollutant might only mean slight modification of its chemical structure, that does not suppress its toxicity. This is a concern, in particular because mild singlet oxygen is present. Although the process is interesting is still far away from real applications (should these materials be kept in suspension? supported? at what concentration? how can it be removed after reaction? is cobalt leaching becoming more relevant after longer treatment times?). This should be indicated in this section, as with available data, comparisons with other processes should not be done.

Reviewer #3 (Remarks to the Author):

In this study, the authors reported an interesting work on the nanoconfinement-enabled Fenton-like system. They observed that Co₃O₄ nanoparticles confined in CNT exhibited much higher activity to activate PAA and other peroxides than the unconfined analogue, and the underlying mechanism responsible for such nanoconfinement effect were probed based on collective experiments and DFT calculation. After systematic comparison study and analysis of the reaction products, they concluded that singlet oxygen should be the main active species for BPA degradation. Also, they proposed a mass transfer-chemical kinetic model to simulate the diffusion and reaction of the active species toward the target compound. Generally, it provides a successful case to demonstrate the nanoconfinement effect on catalytic water treatment. However, after carefully going through the whole manuscript and checking the key references, I think the novelty of this submission is relatively weak since some previous studies have reported such similar and unique nanoconfinement effect (Yang et al., PNAS 2019; Zhang et al., Nature Commun 2022, both cited in this study). Particularly, in Yang's work they used Fe₂O₃ nanoparticles embedded inside CNT (Co₃O₄ used in this study) and reported a singlet oxygen-mediated Fenton-like reaction (similar to this study but the authors did not mention it in the introduction section, a little surprised). In addition, I have other comments on this work:

L 93-95 "Theoretical studies have evidenced that the concave surface is more feasible for adsorption than the convex surface, thus the reactive species are prone to enrich on the interior surfaces" Please check if it is right for all the reactive species or just suitable for some specific species?

L 109-122. The authors are advised to carefully address the advantage of PAA. Is it economically available? In addition, one should be specified that using PAA means the addition of more carbon sources to the target water. This is why inorganic peroxide is widely used instead of the organic one.

Figure 1. L 132 "in" should be "out". The spatial structure of the hybrid catalysts should be further demonstrated by using elemental mapping. The authors provided some relevant results in the supplementary information, however, they did not specify the mapping region and one cannot get what does Figure S1 mean. I think it is very important to support the nanoconfined structure of the target catalyst, and obviously, the current version did not provide very solid evidence to support their claim in L145-149. In addition, as for Figure 1-f,g, how many particles were considered for size distribution analysis? The authors are advised to specify the details of the pristine CNT structure, including the pore size and surface chemistry.

L161-163 "an obvious increase at P/P₀ > 0.8" I did not observe that in Figure S2. In addition, since the unit of N₂ adsorption is cm³/g, do both catalysts have similar weight? I am also surprised that both adsorption curves overlap well, does the metal confinement not affect the pore structure of CNT support? It seems out of expectation.

L 199-201 the authors should examine why the catalytic performance drop dramatically for the "out" sample? All resulting from Co leaching or any other reasons? Also, I am a little curious if the reaction products will be adsorbed by CNT sample and if yes, does the adsorption affect the catalytic activity??

L 208-210 I think such comparison is of limited significance since the conditions varied greatly. In addition, the text for mechanism discussion (including species analysis, DFT, model development and analysis) is somewhat lengthy and should be more concise.

For SI

Table S2 is of limited significance

Figure S1 Mapping regions should be specified

Figure S3 Sample names? (in or out?)

Figure S7 Why leaching will reach equilibrium? More explanation is required

Figure S11 delete "in"

Figure S23-b absorption or adsorption? Check it in the y axis

Reviewer #4 (Remarks to the Author):

This paper intends to facilitate mass transfer in electro-Fenton systems by encapsulating the catalyst inside carbon nanotubes. In this case, the selected catalyst Co_3O_4 is not well justified. There are also minor flaws such as ageing experiments not pushed hard enough (10 cycles of electro-Fenton degradation and 150 min, line 200-202). But the main drawback is the excessive length of the various mechanism sections: 16 pages in total + 3 very convoluted figures. I am lost and I do not have the patience to dedicate the time that I would need to decipher those. `

Journal: *Nature Communications*

Manuscript ID: NCOMMS-22-45381

Title: **“Tuning of Fenton-like Reactions from Radicals to Nonradical Process for Selective Water Decontamination under Spatial Nanoconfinement”**

Author(s): Tongcai Liu, Shaoze Xiao, Nan Li, Jiabin Chen, Xuefei Zhou, Yajie Qian, Ching-Hua Huang, Yalei Zhang

Note:

- Our responses to the reviewers' comments are shown in color **blue**.
- All actions of editing taken are shown in color **red**.
- The line and page numbers referred correspond to those in the revised manuscript.

REVIEWER COMMENTS

Reviewer #1 (Remarks to the Author):

This manuscript reported a very interesting and important work on the development of advanced oxidation processes for organic pollutant degradation, which uses the nanoconfinement effect to encapsulate pollutants and reactive species within the critical diffusion length to reduce the mass transfer limitation and improve the utilization rate of reactive species. Since there are numerous studies reporting development of heterogenous catalysts to promote Fenton-like reactions, this work takes a novel perspective on key scientific questions. Intriguingly, the effective regulation of the reaction mechanism of heterogeneous catalytic oxidation system was realized under spatial nanoconfinement, which provides a brand-new strategy for the transformation of reaction pathway from radical to nonradical in the heterogeneous Fenton-like reaction. In addition, the authors have provided a lot of fresh knowledge on the nanoconfinement effect and mass transfer models. Nonradical dominated

*oxidation systems show strong selectivity, and the nano-confined catalyst presents shell layer (size exclusion) and core/shell (confinement effect) further enhance the selectivity of 1O_2 towards water purification, which greatly solves the practical application issues in water treatment. The manuscript is well written and the new mechanisms and phenomena are well verified by a complete and logical experiments. Both in terms of method development and mechanism interpretation, this work has obtained meaningful results in water purification. Consequently, I am sure that this work deserves publication in Nature Communications after **minor revision** according to the following comments:*

Response: We are grateful to the reviewer for your comments. Your insightful comments will greatly help to improve the quality of this paper, and our detailed response to your comments is as follows.

1. The authors make an exciting statement on the mechanism of peracetic acid (PAA)-based pollutants removal systems, yet the experimental results were significantly different from current reports in the PAA-based oxidation system (a process dominated by free radicals). Considerable efforts have been made to explore the mechanism of the formation of active species in the manuscript. I'm curious about the advantages that nonradicals have over radical oxidation systems in water treatment.

Response: Thanks for the reviewer's comment. Typically, the oxidation capacities of the nonradical systems are moderate compared with the highly oxidizing radical systems, resulting in insufficient mineralization of organic pollutants. Nevertheless, nonradical oxidation has a few technical advantages over radical oxidation in actual water treatment. For instances, (i) nonradical systems exhibit high selectivity toward electron-rich organic substances and bacteria; (ii) nonradical oxidation can maintain its excellent efficiency in complicated water matrices in a wide pH working window, despite the presence of inorganic anions and NOM; (iii) the nonradical system would

not produce toxic halogen-containing products (e.g., ClO_3^- , BrO_3^- , chlorinated and bromated disinfection byproducts) even at a high concentration of halide ions; (iv) nonradical pathways have a high peroxide utilization efficiency with low stoichiometry between the consumed persulfate and target contaminants, which remarkably reduces chemical inputs and is more economical compared with conventional AOPs; and (v) the redox potential of nonradical pathways is tunable to be regulated by structure/composition (e.g., heteroatoms and defects) and physicochemical properties (e.g., electrical conductivity and Zeta potential) of the catalysts ¹. The comment was also addressed by major comment 2 of Reviewer 2, please refer to our response there for detailed discussion.

Action: We have the added relevant statements in the revised Supporting information (Text S21).

“Overall, nonradical pathways (e.g., $^1\text{O}_2$) has several advantages, including: a) making full utilization of oxidants' oxidizing capacities; b) avoiding the self-quenching of radicals produced during the reaction; and c) minimizing interference from organic and inorganic compounds in the real water environment. In addition to the abovementioned advantages, the selective (mild) oxidation capacity inevitably has some drawbacks. Specifically, nonradicals with mild oxidation capabilities have comparatively slow degradation rate and low mineralization rates for the target organic contaminants. Hence, appropriate strategies should be employed to maximize the function of nonradical oxidation mechanism in practical applications. Firstly, the oxidation of refractory pollutants has been evidenced as a good method to improve the biodegradability, implying that oxidation would be integrated with the subsequent biotreatment procedure. In comparison with other Fenton-like systems, activated PAA processes would be more appropriate to associate with biotreatment process as the catalytic decomposition of PAA can produce numerous fine carbon sources including methanol, acetic acid and formaldehyde. Hence, combining PAA-based oxidation process and biotreatment process would contain bright potential

in the treatment of contaminated waters, especially when the waters are lack of carbon sources. Secondly, the $^1\text{O}_2$ oxidation process can be employed as pretreatment technique in the wastewater treatment, because $^1\text{O}_2$ can successfully oxidize organic materials into intermediate products with simpler or harmless structures. In addition, the $^1\text{O}_2$ oxidation is inert towards Cl^- compared with the radical oxidation process, thus the nonradical based oxidation process avoid the secondary pollution of harmful by-products (e.g., ClO_3^- and BrO_3^-).

2. *It is very difficult to quantify the mass transfer process and the reactive species in the nanochannel through experiments. The author solves this problem through the model, which I think is a very important method. To validate this model a large amount of experimental data must be contemporary fitted to obtain robust data and to have an idea of the sensitivity of the parameters on the recorded experimental profiles. A carefully evaluation of these points must be carried out, without this critical evaluation the conclusions obtained from the kinetic model are very few supported.*

Response: Thanks for the reviewer's comment. We analyzed the sensitivity of experimental parameters to validate the model. Total S values (eq S29-S31) were used to evaluate the importance of individual reactions for the evolution of major species and the overall kinetics of Fenton-like system. The larger the Total S value, the more important the reaction is to the whole simulation result (Table S11, S12).

$$s_j = \sum_1^n \left(\frac{\frac{k_j^\alpha}{C_i^{\text{exp}}} + \frac{k_j^\beta}{C_i^{\text{exp}}}}{2} \right) \quad (\text{S29})$$

$$S_j = \frac{s_j - \min s_j}{\max s_j - \min s_j} \quad (\text{S30})$$

$$\text{Total } S = \sum_1^m S_j \quad (\text{S31})$$

Where s_j , S_j and $\text{Total } S$ are the sensitivity index, standardized sensitivity index and total standardized sensitivity index, respectively; m is number of reactions; n is the number of species; j is reaction j and range from 1 to m ; α is upper limit of parameter change, 10% or 20%; β is lower limit of parameter change, -10% or -20%;

$C_i^{k_j^\alpha}$ is the modeled value of contaminants concentration timepoint i , with a 10% or 20% increase on k_j ; $C_i^{k_j^\beta}$ is the modeled value of contaminants concentration timepoint i , with a 10% or 20% decrease on k_j .

Action: We have added the sensitivity analysis in the Table S11, 12. In addition, we also have added the related statement in Text S16.

3. The formation of 1O_2 is generally accompanied by the generation of free radicals in Fenton-like systems. In this work, what's the proportion of 1O_2 as an active species in pollutant degradation.

Response: We are grateful to the reviewer for your comments. Generally, the formation of 1O_2 is always accompanied by the production of free radicals such as HO^\bullet and $SO_4^{\bullet-}$. And even the 1O_2 is only a secondary active species. Therefore, there is an urgent need to develop new catalysts for highly selective and efficient generation of 1O_2 . According to the quenching experiment (NaN_3), BPA was almost completely inhibited, indicating that 1O_2 completely acted on BPA oxidation. In addition, mass balance calculation indicated that FFA was oxidatively converted to three classical products with a conversion rate greater than 99% (Fig. 2f), verifying the exclusive role of 1O_2 as the reactive species in the Co_3O_4 -in-CNT/PAA system.

Action: No further action.

4. Lines 554-573: The quenching of radicals in Fenton-like reactions by various inorganic anions or organic compounds limits their value in industrial applications, thus selective oxidation strategy is an important index in practical water decontamination. I think more discussion is needed to improve the selectivity method.

Response: Numerous efforts have been devoted to enhance oxidation efficiency. The

practical application of conventional Fenton-like for water treatment, mainly based on the generation of sulfate radicals ($\text{SO}_4^{\cdot-}$) and/or hydroxyl radicals (HO^{\cdot}). However, the generated reactive oxygen radicals (ROS) would be highly consumed by the co-existed inorganic ions, non-toxic natural organic matter (NOM) and self-quenching, which is severely hindered by poor decontamination selectivity. Thus, to ensure sufficient removal of targeted pollutants (e.g., the emerging micropollutants) among various competitive species in real waters, excessive input of oxidant and/or energy is typically adopted, which not only increases the cost but also may result in undesired by-products. The need to address this challenge has motivated intensive research on selective oxidation technologies via nonradical Fenton-like catalysis. It was reported that High value metals (HVMs) like Cu(III), Fe(IV) and Co(IV) could be formed in AOP systems, which not only showed strong oxidative ability regardless of the co-existing constituents, but also have high selectivity towards electron-donating groups. But the influence of solution pH on the metals form shouldn't be ignored, because it would change the reduction potential of the metals. For instance, the reduction potential of Cu(III)/Cu(II) in solid phase is 2.3 eV, while it decreases to 1.57 eV in ionized form. To enhance selective degradation efficiency, many efforts also have been devoted to yield most $^1\text{O}_2$ and minimum generation of radical. The non-radical $^1\text{O}_2$ is an excited state of O_2 with a short lifetime (1-3 μs) in H_2O . It was known that the lifetime of $^1\text{O}_2$ can be extended in D_2O . The $^1\text{O}_2$ possesses high selectivity, which prefers to attack nucleophiles. Therefore, it is prone to oxidize organic pollutants with electron donating functional groups like $-\text{OH}$, $-\text{NH}_2$ and alkyl group. Moreover, selective oxidation could be achieved based on the differences of the contaminants in electron-donating/withdrawing properties, ionization potential, molecule size, and so on. It is believed that the development of selective oxidation is of greatly significant in water treatment.

In addition, the generated ROS are usually consumed by macromolecular components like NOM, resulting in decrease of degradation efficiency. The co-existed NOM are much larger than organic pollutants such as tetracycline antibiotics, perfluorochemicals and phenols in size. Consequently, they can be intercepted based

on size before oxidation process. Yolk-shell (or core-shell) structure, encapsulating catalysts by a porous shell with specific size, is an ideal strategy to exclude co-existed large molecules. When the shell is much larger than the core, the confinement effect may be formed and significantly promote the reaction kinetics. Yolk-shell structure with cavity between the catalyst (core) and shell is promising not only in selective removal of multi-component contaminants, but also in enriching trace contaminants from aqueous system. Size exclusion is of great significant in practical selective oxidation, since it can reduce the consumption of ROS by the low-toxic compounds in the water like NOM. Consequently, the generated ROS can fully oxidize high-toxicity organic pollutants to low-toxic ones even mineralize them to CO₂. Nevertheless, it is still expected to develop a strategy to anchor and decompose the special functional groups, which might be more efficient for detoxification. In addition, the regeneration of the nano-reactor and the bi-functional membranes/films should be further studied to reduce their cost.

Furthermore, the surface charge of catalysts is mainly determined by their functional groups. Generally, pollutants with counter charge to catalysts would be preferentially adsorbed/enriched and then oxidized. On the contrary, the target pollutants molecules with the same charge would be repulsed from the surface of catalysts, resulting in low removal efficiency. For example, MOFs can be easily modified with diverse functional groups, but few selective oxidation examples by this strategy are reported to date. This maybe because that the electrostatic interaction between the organic pollutants and catalysts is greatly influenced by solution pH, and the adsorption/enrichment process would be codetermined by π - π interaction, hydrogen bonding and so on. In order to strengthen the electrostatic interaction between MOFs and organic pollutants, some anionic/cationic materials were used to prepare composites with MOFs-based catalysts.

Finally, hydrophobicity and hydrophilicity of contaminants exerted great influence on oxidation process in heterogenous catalysis system, which can affect the surface enrichment of organic pollutants on the catalysts. For hydrophobic organic pollutants, they might be enriched near the surface of the hydrophobic catalysts,

which is beneficial for their preferential degradation. Target compounds with larger partition coefficient (XLog P) value, meaning stronger hydrophobicity, exhibited larger adsorption capacity in single contaminant solution. The functional group with larger XLogP value possessed stronger hydrophobicity, leading to faster removal rate. However, the enrichment process of the target contaminant might be affected by the co-existing substances.

Action: We have the added relevant statements in the revised Supporting information (Text S21).

Text S21. Selective oxidation strategies for decontaminants.

Fenton-like oxidation processes are frequently employed to degrade organic pollutants for water purification. However, the quenching of radicals in Fenton-like reactions by various inorganic anions or organic compounds limits their value in industrial applications. The need to address this challenge has motivated intensive research on selective oxidation technologies via nonradical process. Hence, we proposed two pathways to promote selective oxidation for rapid decontamination: tuning the reactive species to enhance selectivity and enhancing mass transfer process to improve kinetics. Specifically, nonradical process (e.g., $^1\text{O}_2$ and high-valent metals) typically exhibit high selectivity toward electron-rich organic substances, and maintain excellent efficiency in complicated water matrices in a wide pH working window, despite the presence of inorganic anions and NOM. In addition, the selective oxidation strategy is to enhance the mass transfer process and improve the kinetics by altering the selectivity of the catalyst. For example, selective contacts between catalysts and contaminants (including size exclusion, surface charge, hydrophobicity). Given the complex components of wastewater, combining various methods for selective oxidation in actual wastewater treatment prove to be promising approach.

5. Some formatting and grammar comments: Line 37: “mass transfer-chemical kinetic model” should be “mass transfer and chemical kinetic model”. Lines 290-291:

this sentence needs to be revised. The expression forms of free radicals need to be unified in the whole manuscript.

Response: Thanks for your kind suggestions and careful work. We have read the manuscript several times and made the corrections.

Action: The above-mentioned errors and other wrong examples have all been corrected and marked in red throughout the whole manuscript.

Reviewer #2 (Remarks to the Author):

*The manuscript NCOMMS-22-45381 deals with a very interesting issue, namely the enhanced removal of pollutants in confined medium. The paper is **well written**, it is easy to follow, results are **interesting** and supported by data. Findings are new and relevant.*

However, I have some comments to do on the work that authors should consider:

Response: Many thanks for your positive comments and encouragement on this work. Also, your insightful comments below are important to improve the quality of this submission and have been carefully addressed. Detailed response to your comments is mentioned below.

1. I am not sure if the word Fenton (or even Fenton-like) describes the process. Fenton deals with decomposition of hydrogen peroxide into highly reactive species by iron salts in a catalytic cycle involving Fe(II) and Fe(III), and where hydroxyl radicals play an important role. Here we have a relevant process, but involving cobalt, peracetic acid (PAA) and seems to be based on singlet oxygen. Although it is related with Fenton chemistry, I suggest changing Fenton-like by cobalt-PAA system, at least in the title.

Response: Thanks for your kind suggestion. In fact, the activation of hydrogen

peroxide (H₂O₂), peroxymonosulfate (PMS) and persulfate (PDS) were also examined in this work. The results collectively indicated that the activation mechanism of peroxides was dominated by radical process (i.e., HO[•] or SO₄^{•-}) in the Co₃O₄-*out*-CNT/peroxides systems, while ¹O₂ play the primary role in the Co₃O₄-*in*-CNT/peroxides systems. Hence, embodying the cobalt-PAA system in the title may limit the paper's readership and diminish its impact.

In addition, we also referred to the titles of several high-level journal papers on persulfate and hydrogen peroxide activation processes. For example, “*Identification of Fenton-like active Cu sites by heteroatom modulation of electronic density, Proc. Natl. Acad. Sci. U.S.A., 2022, 119(8): e2119492119.*”; “*In situ turning defects of exfoliated Ti₃C₂ MXene into Fenton-like catalytic active sites., Proc. Natl. Acad. Sci. U.S.A., 2023, 120(1): e2210211120.*”; “*Self-cycled photo-Fenton-like system based on an artificial leaf with a solar-to-H₂O₂ conversion efficiency of 1.46%, Nat Commun, 2022, 13 (1), 4982.*”; “*Single Cobalt Atoms Anchored on Porous N-Doped Graphene with Dual Reaction Sites for Efficient Fenton-like Catalysis, J. Am. Chem. Soc. 2018, 140, 12469–12475.*”; “*Atomically dispersed golds on degradable zero-valent copper nanocubes augment oxygen driven Fenton-like reaction for effective orthotopic tumor therapy. Nat Commun, 2022, 13(1): 7772.*” Therefore, Fenton-like reactions are an extremely broad concept that involve activation of many oxidants (not just H₂O₂) and various metals as catalysts (not limited to Fe).

Action: No further action.

2. Authors attribute a relevant role to singlet oxygen and this result seems supported. However, this is a weakness of the system, as singlet oxygen is a worse oxidizing agent than hydroxyl radicals, and a wide range of chemical moieties might be reluctant to its action. Hence, I think that it can only be useful for some target molecules.

Response: Thanks for the reviewer's comment. We agree.

Advanced oxidation processes (AOPs) could efficiently eliminate contaminants by the in-situ generated radicals (e.g. hydroxyl (HO^\bullet), sulfate radical ($\text{SO}_4^{\bullet-}$)), which have raised wide attention due to their easy operation, mild reaction conditions and high efficiency. The radicals with high oxidation capacity could not only directly decompose the organic pollutants, but also reduce their toxicity and even mineralize them to CO_2 . Various AOPs, such as photocatalysis, Fenton/Fenton-like reaction, electrocatalysis, ozonation and their coupling technologies have been developed for water treatment in recent decades. **Overall, there are several advantages in exploiting the non-radical pathways (Figure R1):** **a)** full use of the oxidizing capacity of oxidants, **b)** avoiding self-quenching of the generated radicals during the reaction, and **c)** reducing the interference by organic/inorganic compounds in the matrix/environmental background.

Specifically, compared with radical oxidation, nonradical oxidation processes have significant differences in oxidation mechanism, redox potential (relatively mild), evolutionary pathways, and reaction zone (surface region). These differences bring unique characteristics to the nonradical-based catalytic oxidation reaction, mainly reflected in following aspects. **(1) Excellent independence of solution pH.** An example of this is that $^1\text{O}_2$ generated in $\text{Fe}_2\text{O}_3@\text{FCNT-H}/\text{H}_2\text{O}_2$ system can maintain efficient degradation of methylene blue in the range of pH 2.0–9.0. **(2) Strong resistance to versatile inorganic ions and background organic matters,** which universally exist in domestic sewage and industrial wastewaters. For example, Ma et al. found that in NCNTFs/PMS system with nonradical as the main oxidation mechanism, the addition of common anions had almost no effect on the substrate degradation³. **(3) Significant reduction of toxic by-products.** Generally, halogens (Cl^- , Br^- et al.) can potentially be oxidized to form $\text{ClO}_3^-/\text{BrO}_3^-$ in radicals oxidation systems. However, these adverse reactions do not occur in nonradical oxidation processes⁴. **(4) Nonradical oxidation processes can avoid the inefficient consumption of oxidants,** thus improving its stoichiometric efficiency⁴. These characteristics fully indicate that nonradical-based catalytic oxidation reaction has a broad research and application prospect in the field of environmental pollution

treatment.

As discussed above, the nonradical oxidation ($^1\text{O}_2$) processes exhibit moderate redox potential, which enables them to adapt to diverse pH and resist the interference of inorganic ions and background organic matters. Therefore, the nonradical oxidation processes have bright prospects for removing organic pollution from natural water bodies, especially those with high salinity. In addition, due to the need for disinfection and drug therapy, the wastewater produced by medical treatment inevitably contains a large number of organics and Cl^- . Meanwhile, compared with the free radical oxidation processes, these nonradical oxidation processes will not react with Cl^- to produce ClO_3^- , a toxic by-product. Therefore, the nonradical oxidation processes based on heterogeneous catalysis can be used to treat medical wastewater containing antibiotics without worrying about secondary pollution.

On the other hand, the selective (mild) oxidation capacity not only brings the advantages mentioned above, but also comes with some inevitable disadvantages. Such as relatively low mineralization capacity and slow degradation rate for organic matters. Therefore, appropriate strategies should be adopted to maximize the role of nonradical oxidation mechanism in practical applications. For example, $^1\text{O}_2$ process can be used as pretreatment method in the treatment of organic pollution. Although they exhibit limited mineralization capacity, it can effectively oxidize organic matters into intermediate products with simpler structure, which is conducive to the subsequent treatment. Therefore, adding the $^1\text{O}_2$ process before biological oxidation process might be a good countermeasure, which can 1) eliminate the antibacterial properties of antibiotics by destroyed structures, 2) improve the biodegradability, and 3) kill pathogenic bacteria and viruses. Furthermore, postpositional free radicals based oxidation process (or co-existence with $^1\text{O}_2$) can compensate for the shortcomings of biological oxidation, such as deep mineralization of non-biodegradable pollutants and bactericidal effects. Hence, it is also an effective solution to combine nonradical oxidation with free radical oxidation, because the free radicals can oxidize by-products into CO_2 and H_2O more efficiently.

In any case, the potential value of nonradical process for environmental

remediation is always worthy of affirmation. Additionally, it is no doubt that reactive species are the major contributors to the oxidative degradation of organic pollutants. The activation paths in AOPs are sensitive to the reaction conditions, including the physicochemical properties of activating materials, oxidants, contaminants or even the co-existed interfering substances. In addition, how to achieve effective regulation between free radical pathways and non-radical pathways? Is there a threshold between them that can reflect the activation path? These questions remain to be answered.

Figure R1. Comparison of advantages and disadvantages of free radicals and non-free radicals

Action: We have the added relevant statements in the revised Supporting information (Text S21).

“Overall, nonradical pathways (e.g., $^1\text{O}_2$) has several advantages, including: a) making full utilization of oxidants' oxidizing capacities; b) avoiding the self-quenching of radicals produced during the reaction; and c) minimizing interference from organic and inorganic compounds in the real water environment. In addition to the abovementioned advantages, the selective (mild) oxidation capacity inevitably has some drawbacks. Specifically, nonradicals with mild oxidation

capabilities have comparatively slow degradation rate and low mineralization rates for the target organic contaminants. Hence, appropriate strategies should be employed to maximize the function of nonradical oxidation mechanism in practical applications. Firstly, the oxidation of refractory pollutants has been evidenced as a good method to improve the biodegradability, implying that oxidation would be integrated with the subsequent biotreatment procedure. In comparison with other Fenton-like systems, activated PAA processes would be more appropriate to associate with biotreatment process as the catalytic decomposition of PAA can produce numerous fine carbon sources including methanol, acetic acid and formaldehyde. Hence, combining PAA-based oxidation process and biotreatment process would contain bright potential in the treatment of contaminated waters, especially when the waters are lack of carbon sources. Secondly, the $^1\text{O}_2$ oxidation process can be employed as pretreatment technique in the wastewater treatment, because $^1\text{O}_2$ can successfully oxidize organic materials into intermediate products with simpler or harmless structures. In addition, the $^1\text{O}_2$ oxidation is inert towards Cl^- compared with the radical oxidation process, thus the nonradical based oxidation process avoid the secondary pollution of harmful by-products (e.g., ClO_3^- and BrO_3^-).

3. The environmental relevance section should be completely rewritten. In this work only primary removal of pollutants is studied via chromatography. That means that disappearance of the pollutant might only mean slight modification of its chemical structure, that does not suppress its toxicity. This is a concern, in particular because mild singlet oxygen is present. Although the process is interesting is still far away from real applications (should these materials be kept in suspension? supported? at what concentration? how can it be removed after reaction? is cobalt leaching becoming more relevant after longer treatment times?). This should be indicated in this section, as with available data, comparisons with other processes should not be done.

Response: Thank you for the critical comments and helpful constructive suggestions.

1. We agree. In most cases, the research of Fenton-like oxidation focuses on a model pollutant, such as dye, antibiotic, phenol or other pollutants. Decrease of the model organic in concentration (vs time), which is considered as “degradation efficiency (or removal rate)”, is used to judge the oxidation capacity of the system. As we know, the ideal result of oxidation is to neutralize pollutants, whether by mineralization or transformation. However, the degradation of the model organic compound is not equal to harmlessness due to the existence of many toxic degradation intermediates. Therefore, in addition to the parent compounds, it is also crucial to consider the toxicity assessment of intermediates. In order to better describe the transformation of pollutants in the reaction process, we detected the transformation products by HPLC-MS-MS. Moreover, the density functional theory (DFT) calculation was supplemented. For example, the highest occupied molecular orbital (HOMO) energy level analysis and condensed Fukui functions were used to predict the regioselectivity of different reactive species on the molecules, to further verify the reaction mechanism. In addition, the toxicity of BPA and its transformation byproducts was predicted using the USEPA ECOSAR program based on predicted quantitative structure–activity relationship (QSAR). Finally, we determined the acute toxicity of the solution after treatment by measuring the inhibition rate of luminescence against *Vibrio fischeri* bacteria.

$^1\text{O}_2$ species can react selectively with organic contaminants via hydrogen abstraction, oxygen addition, and electron transfer, as well as electrophilic addition. Moreover, $^1\text{O}_2$ also exhibits selective reactivity with organic molecules with a lone pair of electrons or π electrons, such as phenols, aniline, and tertiary amines. Therefore, the reactive sites of BPA were predicted by quantum chemical calculations. The analysis on the highest occupied molecular orbital (HOMO) of BPA indicated that electrophilic attack sites were primarily located on the benzene ring region (Fig. S27). To accurately describe the gain and loss of electrons on each atom, a more accurate condensed Fukui functions (CFF) of electrophilic (f^-) attack was calculated

through the Hirshfeld charge distribution values (Fig. S27). The results showed that O30 and O32 have the highest f^- value and are most vulnerable to electrophilic attack. Comparatively, sites with higher f^- values (C1, C6, C14 and C15) are also potential electrophilic attack sites. The transformation products (TPs) of BPA were analyzed based on the corresponding fragmentation patterns in the MS/MS spectra in the Co_3O_4 -in-CNT/PAA system (Fig. S28-36). As shown in Fig. S37, BPA was oxidized to the hydroxylation product TP244, and the phenolic substituents were oxidized to the benzoquinone product TP256 via one-electron transfer. Subsequently, the benzoquinone structure of TP256 is further oxidized to carboxylic acid products (TP324) by ring-opening reaction. In addition, the C1 or C15 with high f^- values suffered from an electrophilic attack by $^1\text{O}_2$, causing benzene ring cleavage and dicarboxylic acid byproducts formation (TP266 and TP194). Afterwards, the reaction products were further oxidized to TP194 and TP180 via decarboxylation and demethylation. Finally, these intermediates of BPA degradation could further react with $^1\text{O}_2$ and convert to CO_2 and H_2O .

The aquatic toxicity of transformation products was estimated by QSAR analysis using the ECOSAR system. According to the system established by the Globally Harmonized System of Classification and Labeling of Chemicals (GHS), the predicted toxicity values of BPA and all intermediates can be divided into four categories: very toxic ($\text{LC}_{50}/\text{EC}_{50}/\text{ChV} < 1 \text{ mg/L}$), toxic ($1 \text{ mg/L} < \text{LC}_{50}/\text{EC}_{50}/\text{ChV} < 10 \text{ mg/L}$), harmful ($10 \text{ mg/L} < \text{LC}_{50}/\text{EC}_{50}/\text{ChV} < 100 \text{ mg/L}$), and not harmful ($\text{LC}_{50}/\text{EC}_{50}/\text{ChV} > 100 \text{ mg/L}$)⁵. Hence, BPA belongs to the category of acute harmful substances. The results show that all TPs exhibit much lower toxicity than BPA, which was mainly due to the cleavage of the aromatic ring, i.e., the major biotoxic groups (Table S13). The toxicity of each product during the reaction was measured via the above method, and then we determined the acute toxicity of the solution after treatment by measuring the inhibition rate of luminescence against *Vibrio fischeri* bacteria. As shown in Fig. S38, the luminescence inhibition obviously increased to 11.5% in the first 15 min, and the luminescence inhibition reduced 1.5% when the reaction time was further increased to 60 min, which was attributed to the

transformation of the intermediates to other less toxic byproducts. Overall, $^1\text{O}_2$ destruct the bio-toxic moieties in BPA molecules, resulting in the degradation of BPA and the reduction of toxicity.

Action: We have supplemented the discussion about the transformation products and reaction pathways in the revised manuscript (Section 3.7) and supporting information.

“Transformation products (TPs) of BPA were analyzed on an HPLC system (1290uplc, Agilent, USA) connected with a triple quadrupole mass spectrometer (QTOF6550, Agilent, USA), the details are provided in Text S3.” (in lines 557-559)

“The optimized geometry of BPA was optimized using the Gaussian 09 with a basis set of B3LYP/6–31G (d, p) and was visualized using Multiwfn ⁶. The aquatic toxicity of the transformation byproducts was predicted using the USEPA ECOSAR program ⁷.” (in lines 564-567)

“The degradation pathway and toxicity of BPA were further investigated in the Co_3O_4 -in-CNT/PAA system (Text S20). The condensed Fukui functions were calculated to analyze the reaction sites of BPA, with the transformation products (TPs) and degradation pathways proposed in Fig. S37. The EPA’s ECOSAR program was utilized to predict the ecotoxicity of BPA and TPs. The results suggest that the toxicity of TPs was significantly lower than BPA, which was primarily caused by the cleavage of aromatic ring, i.e., one of the key biotoxic groups (Table S13). Moreover, the acute toxicity of the solution after treatment was measured by the inhibition rate of luminescence against *Vibrio fischeri* bacteria. The luminescence inhibition obviously reduced to 1.5% when the reaction time was further increased to 60 min (Fig. S38). Hence, $^1\text{O}_2$ selectively destruct the bio-toxic moieties of BPA and t significantly reduced its toxicity.” (in lines 458-469)

Text S3. Analytical methods of organic compounds.

“The transformation byproducts of BPA were identified by UPLC system

(1290UPLC, Agilent, USA) connected with a triple quadrupole mass spectrometer (QTOF6550, Agilent, USA). BPA was separated on a Zorbax SB-C18 column (2.1 × 150 mm, 5 μm). The mobile phase A was water with formic acid (0.1%, v/v) and the mobile phase B was 100% acetonitrile. The mobile phase B was increased from 20% to 80% in 10 min and kept for 4 min. Afterwards, the mobile phase B decreased from 80% to 20% in 14.1 min and kept for 6 min. MS and MS/MS patterns of BPA and TPs were analyzed with an m/z range of 50 to 500 in negative electrospray ionization (ESI) modes, sheath gas temperature 350°C and sheath gas flow 12 L/min.”

Text S20. Degradation pathway and toxicity analysis.

“The degradation pathway and toxicity of BPA were further investigated in the Co₃O₄-in-CNT/PAA system. The reactive sites of BPA were predicted by the highest occupied molecular orbital (HOMO) and condensed Fukui functions (CFF) of BPA. The results showed that the electrophilic attack sites were primarily located on the benzene ring region, and the sites with higher f^- values (O30, O32, C1, C6, C14 and C15) are potential electrophilic attack sites (Fig. S27). The transformation products (TPs) of BPA were analyzed based on the corresponding fragmentation patterns in the MS/MS spectra (Fig. S28-36). As shown in Fig. S37, BPA is degraded mainly through hydroxylation, ring opening and decarboxylation pathways. Specifically, BPA was oxidized to the hydroxylation product TP244, and the phenolic substituents were oxidized to the benzoquinone product TP256 via one-electron transfer. Subsequently, the benzoquinone structure of TP256 is further oxidized to carboxylic acid products (TP324) by ring-opening reaction. In addition, the C1 or C15 with high f^- values suffered from an electrophilic attack by ¹O₂, causing benzene ring cleavage and dicarboxylic acid byproducts formation (TP266 and TP194). Afterwards, the reaction products were further oxidized to TP194 and TP180 via decarboxylation and demethylation. Finally, these intermediates of BPA degradation could further react with ¹O₂ and convert to CO₂ and H₂O.

The aquatic toxicity of TPs was estimated by QSAR analysis using the ECOSAR system. The results show that all TPs exhibit much lower toxicity than BPA, which

was mainly due to the cleavage of the aromatic ring, i.e., the major biotoxic groups (Table S13). The toxicity of each product during the reaction was measured by the above method, and then we determine the acute toxicity of the solution after treatment by measuring the inhibition rate of luminescence against *Vibrio fischeri* bacteria. As shown in Fig. S38, the luminescence inhibition obviously increased to 11.5% in the first 15 min, and then reduced to 1.5% when the reaction time was further increased to 60 min, which was attributed to the transformation of intermediates to other less toxic byproducts along the reaction. Overall, $^1\text{O}_2$ not only selectively removed water contaminants but also significantly decreased their toxicity.”

Table S8. Cartesian coordinate of BPA for DFT calculations.

Center number	Atom number	Coordinates (Angstroms)		
		X	Y	Z
1	C	-0.107815	1.21103	0.648085
2	C	0.969018	1.542951	-0.185089
3	C	0.895173	2.67472	-1.008234
4	C	-0.255505	3.474571	-0.998201
5	C	-1.332339	3.142651	-0.165026
6	C	-1.258494	2.010881	0.658117
7	H	-0.051433	0.346899	1.276575
8	H	1.847588	0.932249	-0.192746
9	H	-0.311888	4.338703	-1.626689
10	H	-2.210908	3.753353	-0.157367
11	C	3.044691	3.970257	-1.167684
12	C	4.096566	3.427955	-0.417091
13	C	4.97579	4.274971	0.270985
14	C	4.80314	5.66429	0.208468
15	C	3.751267	6.206592	-0.542127
16	C	2.872042	5.359575	-1.230203
17	H	4.228387	2.367179	-0.369356
18	H	5.778919	3.860912	0.844082

19	H	3.619445	7.267367	-0.589859
20	H	2.068911	5.773634	-1.803296
21	C	2.078507	3.039469	-1.923808
22	C	2.821419	1.755818	-2.338444
23	H	2.15011	1.109102	-2.863804
24	H	3.643607	2.009248	-2.97459
25	H	3.186721	1.257218	-1.46503
26	C	1.552747	3.757081	-3.180872
27	H	2.374934	4.010511	-3.817018
28	H	0.881438	3.110365	-3.706231
29	H	1.036568	4.648968	-2.892781
30	O	5.700307	6.528592	0.910589
31	H	6.437972	6.75597	0.339844
32	O	-2.357303	1.672187	1.508296
33	H	-2.959601	1.091956	1.036948

Table S13. Estimated acute and chronic toxicity for fish, daphnid and green algae of BPA and transformation products by ECOSAR.

Compound	Acute toxicity ^a			Chronic toxicity ^a		
	Fish (96-h LC ₅₀)	Daphnid (48-h LC ₅₀)	Green algae (96-h EC ₅₀)	Fish (ChV)	Daphnid (ChV)	Green algae (ChV)
BPA	1.28	5.24	1.33	0.55	1.77	0.227
TP224	2.26	13.1	2.07	1.22	4.57	0.329
TP256	126	73.4	60.1	12.7	7.63	16.6
TP324	1.10E+4	5.85E+3	5.35E+3	995	478	763
TP250	113	66.9	10.2	12.0	8.27	25.5
TP266	297	136	26.3	29.8	15.2	57.5
TP194	90.7	53.3	8.19	9.62	6.57	20.4
TP180	327	131	28.6	31.8	13.8	58.1
TP114	1.34E+4	3.37E+4	6.14E+3	7.24E+3	1.56E+3	1.10E+3

^a Unit = mg·L⁻¹.

Supplementary Fig. 27. (a) Optimized geometry of BPA molecule at the B3LYP/6-31G(d, p) level. (b) HOMO molecular orbitals in optimized geometry of BPA. (c) Hirshfeld charge distribution and condensed Fukui functions of the optimized BPA molecule (N+1 and N-1).

Supplementary Fig. 28. Total ion chromatograms (TIC) of BPA under ESI negative mode with the reaction 0 and 15 min.

Supplementary Fig. 29. The Mass spectrum of TP244 by HPLC-MS/MS (ESI-) and proposed fragmentation pathway.

Supplementary Fig. 30. The Mass spectrum of TP250 by HPLC-MS/MS (ESI-) and proposed fragmentation pathway.

Supplementary Fig. 31. The Mass spectrum of TP266 by HPLC-MS/MS (ESI-) and proposed fragmentation pathway.

Supplementary Fig. 32. The Mass spectrum of TP194 by HPLC-MS/MS (ESI-) and proposed fragmentation pathway.

Supplementary Fig. 33. The Mass spectrum of TP256 by HPLC-MS/MS (ESI-) and proposed fragmentation pathway.

Supplementary Fig. 34. The Mass spectrum of TP324 by HPLC-MS/MS (ESI-) and proposed fragmentation pathway.

Supplementary Fig. 35. The Mass spectrum of TP180 by HPLC-MS/MS (ESI-) and proposed fragmentation pathway.

Supplementary Fig. 36. The Mass spectrum of TP114 by HPLC-MS/MS (ESI-) and proposed fragmentation pathway.

Supplementary Fig. 37. Proposed transformation pathway of BPA in the Co₃O₄-in-CNT/PAA process.

Fig. S38. Evolution of luminescence inhibition along the reaction. [PAA]₀ = 200 μM, [Co₃O₄-in-CNT]₀ = 0.1 g/L, [BPA]₀ = 100 μM, pH = 7.0.

2. As the reviewer exactly said, most nanoconfined catalysts are powder and the problems of efficient separation and related potential toxicity would be encountered in practical applications. At present, the problem of catalyst separation and recovery can be alleviated through the combination of membrane and fixed bed technologies, but whether such a process will affect the activity of the catalyst is worth further investigation. In fact, the stability and catalytical performance of nanoconfined catalysts were dramatically increased owing to the avoidance of the nanoparticles agglomeration, maximizing catalyst exposure resulting from high dispersibility, and the low leakage of metal ions. Furthermore, the conductive carbon scaffolds enable nanocomposites with effective and fast internal electron transfer due to the spatial confinement effect. In addition, the leached Co was remarkably reduced to 0.5 $\mu\text{g/L}$ after reaction for 150 min (10 cycles) in the Co_3O_4 -in-CNT/PAA system (Fig. S7), which was much lower than the health reference level of Co (70 $\mu\text{g/L}$) in drinking water and the recommended limits in reclaimed water (50 $\mu\text{g/L}$) by US EPA.

Action: We have revised related statements in this section. (in lines 530-590)

“The degradation pathway and toxicity of BPA were further investigated in the Co_3O_4 -in-CNT/PAA system (Text S20). The condensed Fukui functions were calculated to analyze the reaction sites of BPA, with the transformation products (TPs) and degradation pathways proposed in Fig. S37. The EPA’s ECOSAR program was utilized to predict the ecotoxicity of BPA and TPs. The results suggest that the toxicity of TPs was significantly lower than BPA, which was primarily caused by the cleavage of aromatic ring, i.e., one of the key biotoxic groups (Table S13). Moreover, the acute toxicity of the solution after treatment was measured by the inhibition rate of luminescence against *Vibrio fischeri* bacteria. The luminescence inhibition obviously reduced to 1.5% when the reaction time was further increased to 60 min (Fig. S38). Hence, $^1\text{O}_2$ selectively destruct the bio-toxic moieties of BPA and significantly reduced its toxicity.

The stability of catalyst is critical factor in practical application. Co_3O_4 -in-CNT

exhibits excellent performance throughout each cycle and the leached Co was 0.5 $\mu\text{g/L}$ after reaction for 150 min, indicating the good reusability. Degradation experiment was further performed in real waters, e.g., tap water, lake water and municipal wastewater, to evaluate the potential environmental application. The excellent degradation efficiency could still achieve, highlighting the practicability of this confined Fenton-like process in water treatment (Fig. 5c). On the one hand, the majority of macromolecular compounds and colloidal water components (e.g., NOM and soluble microbial products) could be excluded by size exclusion, thus the internal space can still serve as catalytic sites in the presence of NOM (Fig. 5a) ^{8, 9}. Additionally, the selective oxidant $^1\text{O}_2$ is less affected by the water matrices than the radicals (Fig. 5b). Overall, the superior performance could be ascribed to the synergetic effects from shell layer (size-exclusion) and core/shell (confinement effect), which further enhanced the selectivity of $^1\text{O}_2$ towards contaminant oxidation in real waters (**Text S21**). Hence, the selective oxidation strategies were proposed, as illustrated in Fig. 5e and Text S21. To further verify the selectivity of the Co_3O_4 -in-CNT catalysis, the removal performances of multiple recalcitrant pollutants of environmental and public health concerns, including pharmaceutical (sulfamethoxazole, SMX), environmental hormone (17α -ethinylestradiol, EE2), dye (Rhodamine B, RhB), pesticide (imidacloprid, IMD) and phenols (4-chlorophenol, 4-CP) were further examined (Fig. 5d). The results confirmed that these pollutants could be efficiently removed under spatial nanoconfinement, while BA, NB, CBZ and ATZ with electron-withdrawing groups were resistant to degradation in the Co_3O_4 -in-CNT/PAA system, suggesting that $^1\text{O}_2$ exhibits a strong selectivity towards pollutants (Fig. S41). Overall, the Co_3O_4 -in-CNT achieves excellent stability and selectivity, and this work expands application of nanoconfined catalyst for water treatment.”

Reviewer #3 (Remarks to the Author):

In this study, the authors reported an interesting work on the

nanoconfinement-enabled Fenton-like system. They observed that Co_3O_4 nanoparticles confined in CNT exhibited much higher activity to activate PAA and other peroxides than the unconfined analogue, and the underlying mechanism responsible for such nanoconfinement effect were probed based on collective experiments and DFT calculation. After systematic comparison study and analysis of the reaction products, they concluded that singlet oxygen should be the main active species for BPA degradation. Also, they proposed a mass transfer-chemical kinetic model to simulate the diffusion and reaction of the active species toward the target compound. Generally, it provides a successful case to demonstrate the nanoconfinement effect on catalytic water treatment. However, after carefully going through the whole manuscript and checking the key references, I think the novelty of this submission is relatively weak since some previous studies have reported such similar and unique nanoconfinement effect (Yang et al., PNAS 2019; Zhang et al., Nature Commun 2022, both cited in this study). Particularly, in Yang's work they used Fe_2O_3 nanoparticles embedded inside CNT (Co_3O_4 used in this study) and reported a singlet oxygen-mediated Fenton-like reaction (similar to this study but the authors did not mention it in the introduction section, a little surprised). In addition, I have other comments on this work:

Response: Thank you for reviewing our manuscript and for the constructive comments, which greatly helped us to improve the manuscript.

1. About Zhang et al.'s and Yang et al.'s works:

The concept of spatial confinement was first introduced by Rothenberger and Gratzel, who witnessed the changes in reaction rate when reactants were confined in a small volume¹⁰. Over the past decades, the expansion of studies on nanomaterials and nanotechnology have generated capacious opportunities to provide further novel approach for a myriad of applications. Yang et al. was the first to apply the concept of nanoconfinement to water treatment. Yang et al. proposed confined Fe_2O_3 into carbon materials to improve Fenton efficiency and pH suitability in view of traditional Fenton's low efficiency and narrow pH application range. Although Yang et al. found

a very interesting and important work on the development of nanoconfined catalysis system for H₂O₂ activation, the concept has not been clearly defined and the underlying mechanisms have not been well recognized, specially how to regulate the reaction pathway, reaction kinetics and selectivity are still unclear and yet to be revealed. For example, as Yang et al.'s work described that: “Although we have our preferential guess of the nanoconfinement based on the adsorption-dependent process, we do not have solid proof at this point. We would like to leave this question for further exploration.”; “We hope our work could stimulate the researchers to use nanoconfined Fenton's reaction in various applications especially biological systems to realize selective and efficient oxidation.” Therefore, there are still some key knowledge gaps in this field: (1) There has been a knowledge gap of key scientific issues in heterogeneous Fenton reactions, which pose numerous difficulties and uncertainties to employ nanotechnology strategies; (2) Notwithstanding the nanoconfinement effect has been proposed, the concept has not been clearly defined and understood, which is crucial for the directional design of nanoconfined materials and the wide application of the confinement strategy in water treatment; (3) The essential reasons for the significant alters in reaction pathway, reaction kinetics and selectivity need to be experimentally and theoretically interpreted to address the gap in knowledge, which could provide important reference for the development of new nanotechnology under nanoconfinement condition. (4) To reveal the internal relationship between environmental impact and spatial nanoconfinement characteristics, which could bring broad implications for environmental science and technology.

Zhang et al. prepared a two-dimensional cobalt lattice doped titanium oxide (Co-TiO_x) nano-sheet, and assembled it into laminar membrane with 4.6 Å nanoconfined channels, which was successfully applied to activate PMS for the degradation of organic pollutants. Notwithstanding the impressive results, it is apparent that the development and membrane-based nanoconfined heterogenous catalysis systems are still in its inception phase. (1) Since nanoconfinement allows the occurrence of many interesting counterintuitive phenomena that are not observed

under batch mode, **theoretical understandings of the physicochemical features of nanoconfined catalysis supported by experimental findings are required.** For example, what governs the driving force of mass transfer under confinement? What is significance of hydrogen bonding or capillary pressure on the nanofluidic behavior of membranes? How do the electronic interactions occur between catalysts, oxidants and pollutants by membrane-based nanoconfined catalysis? How is the decomposition of oxidant to ROS generation and contribution affected by the extent of nanoconfinement? **(2) It is also crucial that effective regulation of reaction mechanism (i.e., regulation of radicals and nonradicals) and realization of selective oxidation in practical application.** For instance, Zhang et al. reported that the reaction mechanism relies on the strong oxidizing power of radicals ($\text{SO}_4^{\cdot-}$) toward a broad spectrum of recalcitrant pollutants. However, it can cause undesirable catalytic durability due to its high reactivity for both the target compounds and the competing organic/inorganic constituents of water. Meanwhile, the difficulty in the regeneration of active sites due to the energetically unfavorable redox cycles is also considered one of the major drawbacks in their practical application.

Overall, the progress in heterogeneous Fenton-like processes is hampered by several issues including mass transfer limitation, limited diffusion of short-lived reactive oxygen species (ROS), aggregation of nanocatalysts, and loss of nanocatalysts in wastewater treatment. These issues have been addressed in our work by executing the heterogeneous Fenton-like reactions in confinement. Accordingly, **this work aims to address the gap of knowledge by covering four sections including:** (1) to identify the unique properties of nanoconfined catalysis vs conventional heterogeneous catalysis, and provide fundamental understanding of nanoconfinement effect in nanomaterials, which has been largely unclear; (2) to provide novel insights into the mass transfer process occurring at the solid-liquid interface, challenging the conventional understanding of heterogeneous catalysis; (3) to quantify the internal mass transfer and the reactive species' lifetimes for the first time by mass transfer and chemical kinetic model; and finally (4) to elucidate that the synergism from shell layer (size exclusion) and core/shell (confinement effect)

protects the internal Co species from metal leaching and improves the selectivity, providing an overview of the selective oxidation strategies roadmap for the future research.

Action: We have cited Yang et al's work in the introduction section, and added related statement:

“In contrast to behavior and reaction pathways in the bulk phase, nanoconfinement alters the interaction of guest ions/molecules with reactive surfaces, leading to altered kinetics of various redox reactions and new requirements for re-understanding surface/interface processes and chemical reactions. For example, Yang et al. reported that singlet oxygen ($^1\text{O}_2$) mediated Fenton-like process under spatial nanoconfinement¹¹. This finding was in stark contrast with the conventional Fenton-like processes, while the underlying reaction mechanism remains to be further revealed, specially how to regulate the reaction pathway, reaction kinetics and selectivity are still largely unclear.” (in lines 78-86)

2. Regarding the reviewer’s concern about novelty, the novelty of this work can be summarized below, and the revision actions taken to better reflect the novelty in the manuscript are also indicated.

Novelty and new contributions of this work:

(1). Most of the previous research efforts are focused on the design and synthesis of catalysts that improve the efficiency of peroxides activation and radical’s production in order to overcome the limitations of heterogeneous Fenton processes. **However, there has been a significant gap of key scientific issues in recent years.** The limitation in mass transfer of short-lived radical species is an inherent drawback of the heterogeneous system. The extremely short lifetime of radicals in the aqueous phase ($< 10 \mu\text{s}$) hinders its mass transfer from the generation site (i.e., the catalyst surface) to the bulk solution, which severely limits the effective utilization of radicals. The current study addressed this gap.

Action: Related statements are:

“To date, tremendous efforts have been devoted to design nanocatalysts and to

elaborate the reaction characteristics (e.g., kinetics and mechanisms) of heterogeneous Fenton-like systems. Nevertheless, the mass transfer limitation is the inherent defect of heterogeneous reactions, yet this issue has been largely overlooked in Fenton-like reactions. The extremely short-lived free radicals (10^{-6} - 10^{-9} s) hinder their mass transfer from the generation sites on the catalyst surface to the target pollutants in bulk solution, severely limiting their utilization in the heterogeneous reactions. Moreover, radicals would be inevitably consumed by the water matrices (e.g., dissolved organic matter and coexisting inorganic ions), significantly reducing their reaction with target pollutants. Consequently, there is an urgent need for new nanotechnologies to improve the efficient utilization of radicals in heterogeneous Fenton-like systems.” (in lines 52-63)

(2). This study provides new evidence to disclose nanoconfinement effect in Fenton-like reactions. Yang et al. discovered the excellent properties of nanoconfinement in Fenton reaction, but the scientific problems of heterogeneous Fenton reaction have not been elaborated. Inspired by this work, we propose nanoconfinement strategy to solve the inherent problems in heterogeneous Fenton-like. Encapsulation of short-lived radical species and reactants within the critical diffusion length scale through spatial confinement provide a novel and promising strategy to overcome the challenges in heterogeneous Fenton-like reactions. In addition, Yang’s work did not provide a deep explanation for the nanoconfinement phenomenon, and leave this question for further exploration. Therefore, this novel Fenton-like reaction mechanism under spatial nanoconfinement was elucidated from the perspectives of thermodynamics and kinetics in our work. Overall, enrichment of reactants, more efficient diffusion and reactions, and stronger electronic contacts at nanoscales were all interpreted both experimentally and conceptually in order to better reveal the nanoconfinement catalysis effect in Fenton-like reaction. Specifically, spatial confinement speeds up mass and heat transfer by reducing the reactant diffusion distance. It also improves covalent and noncovalent interactions between the reactant molecules and the reaction surface, particularly for those reactions depending

on distance, such electrostatic contact. Moreover, spatial confinement also favors their individual activity, such as accumulation, dispersion and binding, by promoting interaction, which help to enrich the reactants and thus **thermodynamically supports** the oxidation reaction.

In addition to reactants enrichment, it is possible that the **electronic transport characteristics could be altered obviously, which might potentially lead to quantum mutation**. Herein, density functional theory (DFT) calculations suggested that the confined ultrasmall space limits the electronic motion, then alters the electronic properties and structure, which lead to quantum mutation, and change the transition state to lower the activation energy barriers. Additionally, charges redistribution on the inner surface would result from spatial confinement, which will change the reaction state of the active sites and benefit the accumulation of excess charges around the active sites, leading to the enhanced adsorption and oxidation reaction. On the one hand, the confined space increases the enriched reactive species' utilization efficiency. For instance, the environment's abundant natural organic matter (NOM), as well as bicarbonate, carbonate, and other chemicals, readily quench the generated reactive species, giving them ultra-short lives (10^{-6} - 10^{-9} s) and short transfer distances. On the other hand, the energy barriers of oxidation reaction could be reduced under spatial confinement. Spatial confinement also increases the activity of the electron donor/receptor and accelerates electron transport by regulating the surface electron properties. These enable spatial confinement to **enhance the kinetics** of the reaction to produce an effective oxidation process.

Action: Related statements are:

“In contrast to reaction behavior and pathways in the bulk phase, nanoconfinement alters the interaction of guest ions/molecules with reactive surfaces, leading to altered kinetics of various redox reactions and new requirements for re-understanding surface/interface processes and chemical reactions. For example, Yang et al. reported that singlet oxygen ($^1\text{O}_2$) mediated Fenton-like process under spatial nanoconfinement ¹¹. This finding was in stark contrast with the conventional Fenton-like processes, while the underlying reaction mechanism remains to be further

revealed, specially how to regulate the reaction pathway, reaction kinetics and selectivity are still largely unclear.” (in lines 78-86)

Some related statements are already present in the manuscript: including the section “Theoretical Study on the Regulation Mechanism” and “Mechanistic Insight into Kinetics Enhancement under Spatial Nanoconfinement”. These discussions provide new evidence to disclose nanoconfinement effect in Fenton-like reactions.

(3). The nano-scaled interlayer spacing can greatly enhance the interaction between the ROS and target pollutants with more effective diffusion and reactions, which can also significantly accelerate the catalytic process. However, **qualitatively and quantitatively simulating the molecular mass transfer process at the nanoscale is exceedingly complex and challenging. We have innovatively developed a model for the first time to simulate the mass transfer process in the confinement effect.**

Consequently, a new heterogeneous catalytic model including fluid dynamics, mass transfer and chemical reaction kinetics was developed to better understand the spatial distribution of ROS and mass transfer process. The internal mass transfer and the reactive species' lifetimes were quantitatively interpreted by mass transfer and chemical kinetic model, which breaks through the traditional understanding of heterogeneous catalysis and provides a new insight into the mass transfer process occurring at the solid-liquid interface. The chemical kinetics and mass transfer coupling approach developed in this work can be further developed to simulate and optimize the mass transfer process in different phases. In particular, it is of great significance to explore the interface behavior mechanism of heterogeneous catalytic reactions.

Action: Related statements are:

“Moreover, a novel mass transfer and chemical kinetic model was developed to determine RS concentration, lifetimes and migration distance along the mass transfer process. The enrichment of guest molecules at Co_3O_4 site greatly reduces the migration distance of $^1\text{O}_2$ (192 nm), and enhances the effective contact between $^1\text{O}_2$ and contaminant within the lifetime of $^1\text{O}_2$ (4 μs), which remarkably enhances the

utilization of $^1\text{O}_2$.” (in lines 36-40)

“Although the heterogeneous Fenton process has been widely applied in water decontamination, the mass transfer process is still poorly understood. In the heterogeneous catalysis, the produced reactive species can be localized to the catalyst surface (e.g., activated peroxide complexes, and surface-localized radicals), or diffused from the surface in a rather limited distance ¹². Considering the extremely short lifetime and the limited diffusion extent of ROS from the catalyst surface, the oxidation of target compounds potentially occurs in close proximity to the surface rather than in bulk solution ¹³. Consequently, a new heterogeneous catalytic model including fluid dynamics, mass transfer and chemical reaction kinetics was developed to better understand the spatial distribution of ROS and the solid–liquid interface mass transfer process (Fig. 4b).” (in lines 369-406)

(4). The regulation mechanism of Fenton-like reactions from free radical activation to non-radical pathways through nanoconfinement strategy was revealed. At the macro view, spatial confinement enhances reactant enrichment or products to speed up the oxidation reaction, and even modifies the reaction kinetics by altering the reaction pathway. Activation of PAA by catalysts or external energy is deemed as an effective strategy to strength the efficiency of pollutants removal. Previous reported that the enhanced oxidizing property is mainly derived from its generation of highly reactive radicals, such as acetylperoxy ($\text{CH}_3\text{CO}_3^\bullet$), acetoxy ($\text{CH}_3\text{CO}_2^\bullet$), and hydroxyl radicals (HO^\bullet). However, **the redox reaction energy diagram and reaction pathway are substantially altered** by interactions between nanoconfined space and either oxidizing agent, catalysts or organics. In this work, we demonstrated that nonradical ($^1\text{O}_2$) rather than conventional organic radicals (R-O^\bullet) in PAA activation, played major roles in contaminant degradation. Nevertheless, radical (e.g., R-O^\bullet) mechanism is still the dominant pathway of pollutant degradation in the unconfined system. Herein, we confirmed that Co_3O_4 -in-CNT/PAA reaction enables the generation of $^1\text{O}_2$ based on multiple lines of evidences, including FFA based probe experiments, EPR and isotope technique, and electrochemical experiment. In-depth

investigation reveals the regulation mechanism of the Fenton-like reaction from radical activation to nonradical pathway through nanoconfinement strategy.

Action: There are relevant statements in the manuscript, for instance, reactive species analysis from the apparent phenomenon of radical to nonradical pathway change under spatial nanoconfinement (pages 11-16); The DFT calculation indicates thermodynamically that the nanoconfined space limits the movement of electrons, thus changing the property and structure of electrons, leading to quantum mutation, changing the transition state to reduce the activation energy barrier and promoting the generation of $^1\text{O}_2$ (pages 18-19).

(5). Reveal the essence of selective enhancement or alteration. Fenton-like oxidation processes are frequently employed to degrade organic pollutants for water purification. However, the quenching of radicals in Fenton-like reactions by various inorganic anions or organic compounds limits their efficiency in practical applications. Thus, to ensure sufficient removal of targeted pollutants among various competitive species in real waters, excessive input of oxidant and/or energy is typically adopted, which not only increases the cost but also may result in undesired by-products. This challenge has motivated intensive research on selective oxidation technologies via nonradical Fenton-like catalysis. The selectivity can be significantly accelerated under mild process conditions by confining catalytic reactions within a nanoscale space, while the essence of selective enhancement or alteration is not clear. Hence, we proposed two excellent selective oxidation approaches to achieve rapid decontamination in confined system: tuning the reactive species to enhance selectivity and enhancing mass transfer process to improve kinetics. Specifically, nonradical processes (e.g., $^1\text{O}_2$) typically exhibit high selectivity toward electron-rich organic substances, and maintain excellent efficiency in complicated water matrices in a wide pH working window, despite the presence of inorganic anions and NOM. In addition, the selective oxidation strategy is to enhance the mass transfer process and improve the kinetics by altering the selectivity of the catalyst, such as selective contacts between catalysts and contaminants (including size exclusion, surface charge,

hydrophobicity). Hence, the synergism from shell layer (size exclusion) and core/shell (confinement effect) protects the internal Co species from metal leaching and improves the selectivity, which solves the practical application issues in water treatment.

Action: We have added related statements in revised manuscript and supporting information (Text S19, 21).

“The excellent degradation efficiency could still achieve, highlighting the practicability of this confined Fenton-like process in water treatment (Fig. 5c). On the one hand, the majority of macromolecular compounds and colloidal water components (e.g., NOM and soluble microbial products) could be excluded by size exclusion, thus the internal space can still serve as catalytic sites in the presence of NOM (Fig. 5a)^{8,9}. Additionally, the selective oxidant $^1\text{O}_2$ is less affected by the water matrices than the radicals (Fig. 5b). Overall, the superior performance could be ascribed to the synergetic effects from shell layer (size-exclusion) and core/shell (confinement effect), which further enhanced the selectivity of $^1\text{O}_2$ towards contaminant oxidation in real waters (**Text S21**). Hence, the selective oxidation strategies were proposed, as illustrated in Fig. 5e and Text S21. To further verify the selectivity of the Co_3O_4 -in-CNT catalysis, the removal performances of multiple recalcitrant pollutants of environmental and public health concerns, including pharmaceutical (sulfamethoxazole, SMX), environmental hormone (17α -ethinylestradiol, EE2), dye (Rhodamine B, RhB), pesticide (imidacloprid, IMD) and phenols (4-chlorophenol, 4-CP) were further examined (Fig. 5d). The results confirmed that these pollutants could be efficiently removed under spatial nanoconfinement, while BA, NB, CBZ and ATZ with electron-withdrawing groups were resistant to degradation in the Co_3O_4 -in-CNT/PAA system, suggesting that $^1\text{O}_2$ exhibits a strong selectivity towards pollutants (Fig. S41). Overall, the Co_3O_4 -in-CNT achieves excellent stability and selectivity, and this work expands application of nanoconfined catalyst for water treatment.” (lines 474-495)

Text S21. Selective oxidation strategies for decontaminants.

“Fenton-like oxidation processes are frequently employed to degrade organic pollutants for water purification. However, the quenching of radicals in Fenton-like reactions by various inorganic anions or organic compounds limits their value in industrial applications. The need to address this challenge has motivated intensive research on selective oxidation technologies via nonradical process. Hence, we proposed two pathways to promote selective oxidation for rapid decontamination: tuning the reactive species to enhance selectivity and enhancing mass transfer process to improve kinetics. Specifically, nonradical process (e.g., $^1\text{O}_2$ and high-valent metals) typically exhibit high selectivity toward electron-rich organic substances, and maintain excellent efficiency in complicated water matrices in a wide pH working window, despite the presence of inorganic anions and NOM. In addition, the selective oxidation strategy is to enhance the mass transfer process and improve the kinetics by altering the selectivity of the catalyst. For example, selective contacts between catalysts and contaminants (including size exclusion, surface charge, hydrophobicity). Given the complex components of wastewater, combining various methods for selective oxidation in actual wastewater treatment prove to be promising approach.”

L 93-95 “Theoretical studies have evidenced that the concave surface is more feasible for adsorption than the convex surface, thus the reactive species are prone to enrich on the interior surfaces” Please check if it is right for all the reactive species or just suitable for some specific species?

Response: We agree. Thanks for your careful and rigorous work. I'm terribly sorry for our unclear statement. Specifically, the stress variation of the curved carbon shell endows a deviation of graphene layers from planarity, leading to a shift of p-electron density from the internal surface (concave) to the external surface (convex), which correspondingly results in an electron-deficient and an electron-enriched state to the interior and exterior positions, respectively. Therefore, the hollow-nanostructured materials tend to adsorb more electron-rich reactant molecules, which concentrate

inside hollow structures, thus accelerating the reaction rate. The changes in the manuscript are highlighted in red.

Action: We have revised the related statements:

“Theoretical studies have evidenced that the concave surface is more feasible for adsorption than the convex surface, and some electron-rich reactant molecules are prone to enrich on the interior surfaces.” (In lines 93-94)

L 109-122. The authors are advised to carefully address the advantage of PAA. Is it economically available? In addition, one should be specified that using PAA means the addition of more carbon sources to the target water. This is why inorganic peroxide is widely used instead of the organic one.

Response: We thank the reviewer for pointing out this issue.

1. The advantage of PAA: Peracetic acid [PAA, $\text{CH}_3\text{C}(\text{O})\text{OOH}$] has been extensively studied and utilized in various industries since the formation of PAA was first reported in 1902¹⁴. Afterwards, PAA is a widely used oxidant/disinfectant in various industries, including medicine, food processing, textile, and pulp and paper industries. PAA was first registered in the United States as a pesticide for usage as a disinfectant, sanitizer, and sterilant in 1985¹⁵. On the basis of the evaluation by the U.S. Environmental Protection Agency (USEPA) in 1993, PAA has been used as a disinfectant for various types of equipment and facilities in the United States¹⁶. The European Commission (EC) approved PAA as an existing active substance for use in biocidal products in 2016¹⁷. The Canadian General Standards Board Permitted Substances List (CAN/CGSB-32.311-2015) permits the use of PAA as a food grade cleaner, disinfectant, and sanitizer. In addition, PAA has been used as a bleaching agent because PAA can selectively remove the lignin from pulp¹⁸. PAA might also have the potential to replace several other industrial biocides with undesirable properties, such as formaldehyde, bromine, or isothiazoline. Furthermore, PAA have many qualities of an ideal disinfectant, such as, toxicity to microorganisms, but not to higher forms of

life; effectivity at ambient temperatures; stability and long shelf life; low corrosivity; deodorizing ability; widespread availability; and reasonable cost ¹⁹.

However, the interest towards the water and wastewater treatment applications emerged much later for PAA in the late 1970s and early 1980s ²⁰. The application of PAA to municipal wastewater treatment for reuse in agriculture was first evaluated in a research project approved in 1995 by the EC. One of the main drivers in this development was the increased awareness of disinfection by-products (DBPs) resulting from the use of chlorine compounds. In Canada and parts of Europe, chlorine is not allowed to be used to disinfect wastewater effluents because of its potential to generate harmful DBPs; instead, they recommend PAA ²¹. In the United States, PAA disinfection has been applied in a number of full-scale wastewater treatment plants (WWTPs), and the USEPA has recommended PAA as a disinfectant for use in combined sewer overflows ²². One advantage of PAA is an easy retrofit for sodium hypochlorite disinfection equipment that is present in many existing WWTPs, thus reducing costs for the switch ²³. Residual PAA readily decomposes into harmless compounds such as acetic acid and oxygen. Thus, no quenching of residual is required with PAA disinfection and oxidation. Notably, PAA has been applied in full-scale wastewater treatment plants (WWTPs) in the United States, Canada, and parts of Europe. In 2013, the amount of PAA used in water treatment on a global scale was approximately 29.01 kt and this has been estimated to steadily increase. The global PAA market was worth \$650 million in 2017 and is projected to grow to \$1.3 billion by 2026, including a steady annual 8% increase in wastewater treatment usage ²⁴. This illustrates that PAA has become an accepted alternative oxidant/disinfectant chemical, especially in wastewater disinfection.

More recently, PAA has been studied for the degradation of harmful micropollutants (MPs) in wastewater. In particular, activation of PAA by energy (UV or solar) or catalysts [e.g., Fe(II/III), Co(II/III), Mn(II/III/IV), Ru(III)], and activated carbon has shown high efficiency for the removal of a wide range of MPs ¹⁵. The enhanced removal of pollutant is derived from the formation of organic radicals (e.g., $\text{CH}_3\text{C}(\text{O})\text{O}^\bullet$ and $\text{CH}_3\text{C}(\text{O})\text{OO}^\bullet$). These organic radicals have a longer half-life time

and higher selectivity in degrading aromatic pollutants compared to inorganic radicals (e.g., HO[•] and SO₄^{•-})²⁵. The formation of organic radicals is advantageous because organic radicals are less susceptible to scavengers compared to radicals such as HO[•] and thus have a longer lifetime in waters containing a variety of constituents to oxidize contaminants. In addition, PAA has recently been identified as a promising candidate oxidant owing to it has a high redox potential of 1.96 V and an O=O bond binding energy of 159 kJ mol⁻¹, which is lower than that of conventional oxidants²⁵.

Overall, PAA has been widely applied in various industries, with wastewater treatment applications being **primarily driven** by the growing awareness of DBPs brought on by the usage of chlorine compounds. Therefore, a **significant advantage** of PAA disinfection over chlorine disinfection is the absence of toxic residuals (acetic and oxygen) and DBPs. **Another advantage** of PAA is that it has a high redox potential of 1.96V and O=O binding energy of 159 kJ mol⁻¹, which is lower than conventional oxidants and more easily activated, so it is considered as an attractive oxidant option. **Finally**, the activation of PAA produces organic radicals (R-O[•]) with longer half-lives and unique contaminant selectivity, which facilitates mass transfer in Fenton-like processes and is advantageous for the practical application of PAA.

Action: Related statements are:

“PAA is an emerging oxidant and disinfectant for wastewater treatment due to limited harmful disinfection byproduct formation. PAA based Fenton-like systems have received increasing attention in degradation of refractory pollutants owing to the generation of powerful organic radicals (R-O[•]), e.g., acetoxy(per) radicals (CH₃C(O)O[•] and CH₃C(O)OO[•]). Furthermore, the R-O[•] possess longer half-life and exceptional contaminants selectivity compared to inorganic radicals, rendering the mass transfer effortless in Fenton-like process. Hence, PAA was selected as the oxidant for the in-depth exploration of the spatial nanoconfinement in Fenton-like reactions.” (in lines 108-114)

2. Economic feasibility of PAA: Analysis of the cost of PAA-based disinfection and oxidation is not straightforward as it depends on several case-specific factors, such as

the required microbial quality, the availability of chemicals and the physicochemical properties of wastewater. PAA is less expensive than conventional oxidants (e.g., persulfates), but PAA is liquids with additional costs associated with storage and transportation. Hydrogen peroxide and chlorine are cheaper than PAA, but H_2O_2 is relatively difficult to activate due to its low oxidation performance and high bond energy. Notably, an increase in demand could result in mass-scale production of PAA and subsequent reduction of cost, making it financially comparable with chlorine disinfection. For instance, the amount of PAA used in water treatment on a global scale was approximately 29.01 kt and this has been estimated to steadily increase in 2013. The global PAA market was worth \$650 million in 2017 and is projected to grow to \$1.3 billion by 2026, including a steady annual 8% increase in wastewater treatment usage ²⁴. In addition, chlorine disinfection is efficient but has the risk of DBPs, hence one advantage of PAA is an easy retrofit for chlorine disinfection equipment that is present in many existing WWTPs, thus reducing costs for the switch.

Action: No further action.

3. Introduce more carbon sources when using PAA to treat wastewater.

On the one hand, PAA could self-decompose to produce acetic acid, which would introduce more a carbon source. However, PAA was effective in the disinfection of bacteria and viruses, and the amount of PAA used in the disinfection process is rather low, even lower than hypochlorous acid. In addition, some researchers have combined PAA with other disinfection methods (e.g., UV/PAA) to further reduce PAA dosage and alleviate carbon source introduction ^{26,27}. In this work, PAA is just model oxidant, other inorganic oxidants (i.e., H_2O_2 , PMS and PDS) were also investigated.

On the other hand, the concentration of acetic acid in the treated water can be increased by utilizing PAA in Fenton-like reactions, while the formed acetic acid can be used as a carbon source in the subsequent biological treatment processes. Oxidation of refractory pollutants has been evidenced as a good method to improve

the biodegradability, which suggests oxidation process would be combined with the following biotreatment process. In comparison with other Fenton-like systems, activated PAA processes would be more appropriate to associate with biotreatment process as the catalytic decomposition of PAA can produce numerous fine carbon sources including methanol, acetic acid and formaldehyde. Wang et al. reported that the cell growth of natural planktonic bacteria and *C. vulgaris* after inoculation in the treated sulfamethoxazole solution by the Co/PAA process, respectively. Compared to the experimental results in the absence of PAA, the addition of PAA effectively accelerate the cell growth of both natural planktonic bacteria and *C. vulgaris*, suggesting the good performance in bioavailability when the PAA based Fenton-like process was applied to treat antibiotic wastewater ²⁸. These results illustrated that combining PAA-based oxidation process and biotreatment process would contain bright potential in the treatment of contaminated waters, especially when the waters are lack of carbon sources.

Action: We have added related statements in supporting information (Text S21).

“Firstly, the oxidation of refractory pollutants has been evidenced as a good method to improve the biodegradability, implying that oxidation would be integrated with the subsequent biotreatment procedure. In comparison with other Fenton-like systems, activated PAA processes would be more appropriate to associate with biotreatment process as the catalytic decomposition of PAA can produce numerous fine carbon sources including methanol, acetic acid and formaldehyde. Hence, combining PAA-based oxidation process and biotreatment process would contain bright potential in the treatment of contaminated waters, especially when the waters are lack of carbon sources.”

Figure 1. L 132 “in” should be “out”. The spatial structure of the hybrid catalysts should be further demonstrated by using elemental mapping. The authors provided some relevant results in the supplementary information, however, they did not specify the mapping region and one cannot get what does Figure S1 mean. I think it is very important to support the nanoconfined structure of the target catalyst, and obviously,

the current version did not provide very solid evidence to support their claim in L145-149. In addition, as for Figure 1-f, g, how many particles were considered for size distribution analysis? The authors are advised to specify the details of the pristine CNT structure, including the pore size and surface chemistry.

Response: Thanks for the reviewer's kind suggestions.

1. We have revised "in" to "out" in line 132.
2. We agree. The spatial structure and element composition of catalyst need to be further proved by element mapping. Hence, representative high-angle dark-field scanning transmission electron microscopy (HADDF-STEM) images with energy-dispersive X-ray spectroscopy (EDX) elemental mapping was conducted to map the abundance of C, O and Co in the catalysts. And, the specify the mapping region are specified, as shown in Fig. S1.
3. The software Nano Measurer performed statistical analysis on 30–50 particles for the labeling statistics of particle size to produce a statistical report, which was then exported to make graphs.
4. Multiwalled carbon nanotubes (CNTs) ($\langle d \rangle = 10\text{--}20$ nm and $\langle l \rangle = 10\text{--}30$ μm) were purchased from XFNANO Nano (Nanjing, China). We further determined the specific surface area, pore volume and pore size of the pristine CNTs structure by BET test (Fig. S2). The results showed that surface area, pore volume and pore size were 126.7 m^2/g , 0.6187 cm^3/g , and 19.025 nm, respectively. SEM and TEM were also conducted to characterize the morphology and basic structure of CNT materials (Figure R2). In addition, the surface chemistry of the original CNT has actually been measured, such as XRD (Fig. 1), Raman (Fig. S4).

Action: We have added relevant statements in supporting information.

Table S2. Specific surface area of CNT, $\text{Co}_3\text{O}_4\text{-out-CNT}$ and $\text{Co}_3\text{O}_4\text{-in-CNT}$.

Catalysts	$S_{\text{BET}}(\text{N}_2)$ (m^2/g)	Total pore volume (cm^3/g)	Pore diameter (nm)
CNT	126.7	0.6187	19.025
$\text{Co}_3\text{O}_4\text{-out-CNT}$	164.4	0.5645	3.063

Figure S1. (a) SEM images, and (b) TEM images of CNTs.

L161-163 “an obvious increase at $P/P_0 > 0.8$ ” I did not observe that in Figure S2. In addition, since the unit of N_2 adsorption is cm^3/g , do both catalysts have similar weight? I am also surprised that both adsorption curves overlap well, does the metal confinement not affect the pore structure of CNT support? It seems out of expectation.

Response: Thanks for the reviewer’s comment. We redetermined the BET of catalysts Co₃O₄-out-CNT and Co₃O₄-in-CNT with the same weight, and compared the pristine CNTs data (Fig. S2). After the activation treatment of CNTs (acidification and heat treatment), the smooth surface of CNT becomes rough due to the interruption or local destruction, thus making effective utilization of the hollow structure and more surfaces of CNTs. However, its unique hollow structure of CNTs has been preserved. Moreover, the pore volume and diameter of Co₃O₄-in-CNT were decreased, and increased the surface area compared to pristine CNTs (Table S2). Therefore, the metal confinement affects the pore structure of CNTs, which reduced the pore volume and diameter and increased the surface area.

Action: We have revised relevant statements

Supplementary Fig. 2. N₂ adsorption-desorption isotherms.

“The smooth surface of CNT becomes rough during activation treatment (acidification and heat treatment) due to interruption or local destruction, which reduced the pore volume and diameter, and increased the surface area compared to pristine CNTs (Table S2).”

L 199-201 the authors should examine why the catalytic performance drop dramatically for the “out” sample? All resulting from Co leaching or any other reasons? Also, I am a little curious if the reaction products will be adsorbed by CNT sample and if yes, does the adsorption affect the catalytic activity??

Response: We thank the reviewer for pointing out this issue. The adsorption of BPA and intermediate on CNTs may also be one of the reasons for the gradual deactivation of the catalyst in the Co₃O₄-out-CNT/PAA system. To further confirm this conjecture, we supplemented the reusability experiment (each reaction was prolonged to 60 min). As shown in Fig. S8a, the BPA removal efficiency was observed to gradually decrease to 55% after three cycles, which might be ascribed to the coverage of active sites. Heat treatment was further used to remove the surface adsorbates during the regeneration of the catalyst. Results show that the performance of BPA removal can slightly restore via thermal treatment to eliminate the surface adsorbates. Furthermore,

the catalytic activity of the catalyst was determined as a function of the adsorption efficiency of BPA. The result suggested that the increase of catalyst cycles decreased the adsorption efficiency of BPA, but the overall effect of BPA adsorption on its degradation was not obvious ($< 10\%$) (Fig. S8b). Therefore, the adsorption of BPA and its products is one of the factors for the gradual deactivation of catalyst, but is not the dominant one.

Overall, the decrease of catalytic performance can be attributed to several combined effects in the $\text{Co}_3\text{O}_4\text{-out-CNT/PAA}$. (i) The adsorptions of BPA and the intermediates on the catalyst is one of minor reason for the gradual deactivation of catalysts; and (ii) Co ion percolation significantly reduced the activity of catalyst, which had a major impact on the degradation kinetics of pollutants.

Action: We have added the following statements in the revised manuscript, and Fig. S8 has been added in the supporting information.

“Hence, the progressive deactivation of $\text{Co}_3\text{O}_4\text{-out-CNT}$ could be primarily attributed to the considerable leaching of Co, and also slightly contributed from the competitive adsorption of BPA and its intermediates (Fig. S8).” (in lines 189-192)

Supplementary Fig. 8. (a) The evolution of catalytic activity for $\text{Co}_3\text{O}_4\text{-out-CNT}$ in the cycling tests, and the thermal treatment at 932.2 K for 2 h in N_2 was used to regenerate the catalytic activity after the 3rd cycle. **(b)** Relationships between the catalytic activity of $\text{Co}_3\text{O}_4\text{-out-CNT}$ and the effectiveness of BPA adsorption in consecutive runs. $[\text{PAA}]_0 = 200 \mu\text{M}$, $[\text{Co}_3\text{O}_4\text{-out-CNT}]_0 = 0.1 \text{ g/L}$, $[\text{BPA}]_0 = 100 \mu\text{M}$,

pH = 7.0.

“The progressive deactivation of the catalyst in the Co₃O₄-out-CNT/PAA system may also be caused by the adsorption of BPA and intermediate. After three cycles, the BPA removal was seen to steadily decline to 55%, however, the efficacy of BPA removal could be slightly improved by using thermal treatment to remove surface adsorbates. Furthermore, the effectiveness of BPA adsorption was reduced as the number of catalyst cycles increased, but the total impact of BPA adsorption on its degradation was not obvious (< 10%) (Fig. S8b). Hence the adsorption of BPA and its products is one of the factors for the catalyst deactivation, but it is not the dominant one.”

L 208-210 I think such comparison is of limited significance since the conditions varied greatly.

Response: We appreciate your important comments. First of all, we compared the degradation rates of the same pollutant (BPA) in the Fenton-like reactions, and the degradation rate of the pollutant is related to the amount of the catalyst and oxidants, and the concentration of the target pollutant. Generally, the higher the concentration of catalyst and oxidant, the faster the degradation rate of pollutant was obtained. Here, we employ a modified kinetics model (m- k_{obs}) for evaluating the reaction rate of BPA removal efficiency, and the modified k value is referred to the previous work^{29,30}, the formula was described as followed.

$$k_{obs} = \frac{1}{m_{cat}} \times \frac{dC}{dt} \times \frac{1}{m_{oxi}}$$

Where C is the amount of organic pollutants ($\mu\text{mol/L}$), t is the reaction time (min), m_{cat} is the mass of catalysis (g/L) and m_{oxi} is the mass of oxidants (g/L). Therefore, a modified k_{obs} (m- k_{obs}) is applied by dividing the k_{obs} of contaminant degradation with the amount of catalyst addition and oxidants. The k-value of the Co₃O₄-in-CNT/PAA system is 116.12, and this value is also much higher than those reported for other

activators in recent reports.

Action: We have revised relevant statements:

Table S3. The catalytic performance comparison of previously reported Fenton-like catalysts for PMS, H₂O₂ and PAA activation.

Catalyst (g/L)	PMS ^a /H ₂ O ₂ ^b /PAA ^c (g/L)	BPA (μ mol/L)	Removal Efficiency (%)	k_{obs} (min ⁻¹)	k-value ^d	Ref.
DPA-hematite (0.5)	2.0 ^a	65.7	100 (120 min)	0.039	0.039	31
CuFe ₂ O ₄ -Fe ₂ O ₃ (0.2)	0.36 ^a	21.9	100 (10 min)	0.620	8.611	32
Mn _{1.8} Fe _{1.2} O ₄ (0.1)	0.2 ^a	43.8	95 (30 min)	0.102	5.1	33
Fe ³⁺ -g-C ₃ N ₄ (0.1)	0.3 ^a	100.8	100 (15 min)	0.302	10.067	
Fe _{0.8} Co _{2.2} O ₄ (0.1)	0.2 ^a	87.6	95 (60 min)	0.049	2.45	34
Fe ₃ Co ₇ @C (0.1)	0.2 ^a	87.6	95 (30 min)	0.132	6.6	
Fe ₁ Mn ₅ Co ₄ -N@C (0.1)	0.2 ^a	87.6	100 (10 min)	0.480	24	35
Ag/AgCl/Fh (1.0)	0.18 ^b	131.4	100 (60 min)	0.050	0.278	36
Cn-Cu(II)-CuAlO ₂ (1.0)	0.18 ^b	109.6	98 (120 min)	0.030	0.167	37
Ag/AgCl/Fe-S (1.0)	0.1 ^b	43.8	100 (120 min)	0.030	0.3	36
Cu-Al ₂ O ₃ (1.0)	0.2 ^b	87.6	87 (180 min)	0.010	0.05	37
Cu-doped AlPO ₄ (1.0)	0.18 ^b	109.6	92 (180 min)	0.010	0.056	35
S modified Fe ₂ O ₃ (0.2)	0.04 ^b	192.7	100 (20 min)	0.314	39.25	35
Co@N-C-KNO ₃ (0.025)	0.6 ^a	219	100 (10 min)	0.730	48.67	38
Fe-SAC (0.2)	0.4 ^a	109.6	88 (15 min)	0.104	1.30	39
Co-SAC (0.2)	0.4 ^a	109.6	79 (15 min)	0.0832	1.04	39
Mn-SAC (0.2)	0.4 ^a	109.6	70 (15 min)	0.0624	0.78	39
Ni-SAC (0.2)	0.4 ^a	109.6	62 (15 min)	0.052	0.65	39
Cu-SAC (0.2)	0.4 ^a	109.6	56 (15 min)	0.046	0.58	39
FeCo-NC-1 (0.1)	0.2 ^a	87.6	100 (5 min)	1.179	58.95	35
FeCo-NC-2 (0.1)	0.2 ^a	87.6	100 (4 min)	1.252	62.6	35
FeCo-NC-3 (0.1)	0.2 ^a	87.6	95 (6 min)	0.465	23.25	35
2D Fe ₃ O ₄ nanosheets (0.2)	0.5 ^a	87.6	93.2 (10 min)	0.222	2.22	40
Fe ₁ Mn ₅ Co ₄ -N@C (0.1)	0.2 ^a	87.6	100 (10 min)	0.48	24	41
yolk-shell Co/C (0.15)	0.1 ^a	87.6	98 (15 min)	0.5	33.33	42
Co ₃ O ₄ -out-CNT (0.1)	0.0152 ^c	100	52.6 (15 min)	0.049	32.30	This work
Co ₃ O ₄ -in-CNT (0.1)	0.0152 ^c	100	95.8 (15 min)	0.188	116.12	This work

Note: The normalized rate constant was calculated through dividing the reaction rate of pollutant degradation by the catalyst and oxidant concentration.

In addition, the text for mechanism discussion (including species analysis, DFT,

model development and analysis) is somewhat lengthy and should be more concise.

Response: Consequently, the text for mechanism discussion (including species analysis, DFT, model development and analysis) has been partially deleted to make it more concise. The length of the revised manuscript has been greatly reduced, parts of which have been deleted and moved to attachments to make the manuscript more concise. Currently, the manuscript's length stands (i.e., 4909 (text) + 5 figures) excluding title page, abstract, references, methods and figure captions, which meets the Nature Communications word count requirement

Action: We have simplified the relevant description, and the revised parts are marked red in the manuscript.

For SI

Table S2 is of limited significance

Response: Thanks for the reviewer's kind suggestion. We have made this statement in the prior comment.

Figure S1 Mapping regions should be specified

Response: Thanks for the reviewer's careful work. We have annotated the specified mapping region and redrawn Fig. S1.

Figure S3 Sample names? (in or out?)

Response: Thanks for the reviewer's careful work. I'm terribly sorry for our carelessness. The *sample name* is $\text{Co}_3\text{O}_4\text{-in-CNT}$, and the changes in the manuscript are highlighted in red.

Figure S7 Why leaching will reach equilibrium? More explanation is required

Response: Thanks for the reviewer's kind suggestion. In the $\text{Co}_3\text{O}_4\text{-in-CNT/PAA}$ system, the leached Co was always below 0.5 $\mu\text{g/L}$. In the $\text{Co}_3\text{O}_4/\text{PAA}$ and

Co₃O₄-out-CNT/PAA systems, the activity of catalyst decreased obviously with the increase of catalyst cycle times. The ultimate level of Co in the catalyst is rather low since Co ion is continuously precipitated during the reaction process. If the reuse times are increased, the leaking content of Co will only tend to be steady or even decrease until it is entirely precipitated.

Figure S11 delete "in"

Response: Thanks for the reviewer's kind suggestion. We have revised relevant statement.

Figure S23-b absorption or adsorption? Check it in the y axis

Response: Thanks for your careful and rigorous work. In fact, adsorption is what we are trying to express here.

Reviewer #4 (Remarks to the Author):

This paper intends to facilitate mass transfer in electro-Fenton systems by encapsulating the catalyst inside carbon nanotubes. In this case, the selected catalyst Co₃O₄ is not well justified. There are also minor flaws such as ageing experiments not pushed hard enough (10 cycles of electro-Fenton degradation and 150 min, line 200-202). But the main drawback is the excessive length of the various mechanism sections: 16 pages in total + 3 very convoluted figures. I am lost and I do not have the patience to dedicate the time that I would need to decipher those.

Response: Thank you for reviewing our manuscript and for the constructive comments, which greatly helped us to improve the manuscript. We are truly grateful to you for giving us this precious opportunity to revise our paper.

1. Whether the selected catalyst Co₃O₄ is reasonable. Cobalt-based oxides have recently emerged as the optimum candidates for peroxides activation among all transition metal (TM) oxides, benefiting from the abundance of electronic structures, coordination structures, and spin states in cobalt ions.⁴³ Previous studies have

disclosed that the catalytic activities of oxide-type catalysts are highly dependent on the electronic states of TM ions and TM-O covalence, both of which play important roles in the adsorption and activation of peroxides. It has been reported that Co^{2+} ions are capable of donating electrons to peroxides for the production of reactive species, and the high standard reduction potential of $\text{Co}^{3+}/\text{Co}^{2+}$ couple ($E^0 = +1.92$ V vs. NHE) enables efficient self-recycling of Co^{2+} for the catalytic reactions⁴⁴. In addition, the average cobalt concentration in serum and urine of people are approximately $0.1\text{--}0.3$ $\mu\text{g L}^{-1}$ and 0.1 $\mu\text{g L}^{-1}$, respectively⁴⁵. Although cobalt is claimed not to be a hazardous chemical, several studies showed that excessive cobalt ions are possibly toxic and carcinogenic, leading to serious health problems such as asthma, pneumonia and cardiomyopathy. Homogeneous catalytic reactions of Co^{2+} /peroxides system can introduce cobalt ions to water, which is a potential threat to human beings and increases the operation cost due to the loss of cobalt. **Hence, peroxides activation using cobalt containing materials as heterogeneous catalysts seems to be a promising strategy.**

2. About ageing experiments: I apologize for my unclear expression. In fact, this paper is about the application of heterogeneous Fenton-like in water decontamination rather than electro-Fenton. In the electro-Fenton reaction, 10 times of aging is obviously not enough, however, 10 cycles of experiments were shown to be sufficient in the previous studies on the heterogeneous catalytic process. For instance, *Angew. Chem. Int. Ed.* 2021, 60, 22513 – 22521 (**5 cycles**); *Nat. Commun.* 12(1): 4152 (**3 cycles**). *Proc. Natl. Acad. Sci. U.S.A.*, 2022, 119(31): e2201607119 (**5 cycles**).

3. About the length of the manuscript:

(1) In the mechanism analysis section, our writing logic is as follows: from the identification and production mechanism of reactive species, to the theoretical analysis of mechanism regulation, followed by the theoretical analysis of kinetic improvement, and finally to the environmental application. In fact, the description of mechanism section in our work is also referred to a number of other high-level journal

papers. For example, *Proc. Natl. Acad. Sci. U.S.A.*, 2022, 119(31): e2201607119; *Proc. Natl. Acad. Sci. U.S.A.*, 2022, 119(8): e2119492119; *Nat. Commun.*, 2021, 12(1): 3508; *J. Am. Chem. Soc.* 2018, 140, 12469–12475. Hence, the logic of this work is not problematic, but it needs to be further simplified.

(2) Consequently, the text for mechanism discussion (including species analysis, DFT, model development and analysis) has been partially deleted to make it more concise. The length of the revised manuscript has been greatly reduced, parts of which have been deleted and moved to attachments to make the manuscript more concise. Currently, the manuscript's length stands (i.e., 4909 (text) + 5 figures) excluding title page, abstract, references, methods and figure captions, which meets the Nature Communications word count requirement

Reference

1. Shao P, et al. Potential Difference Driving Electron Transfer via Defective Carbon Nanotubes toward Selective Oxidation of Organic Micropollutants. *Environ. Sci. Technol.* **54**, 8464-8472 (2020).
2. Li Z-Y, Wang L, Liu Y-L, Zhao Q, Ma J. Unraveling the interaction of hydroxylamine and Fe (III) in Fe (II)/Persulfate system: A kinetic and simulating study. *Water Res.* **168**, 115093 (2020).
3. Ma W, Wang N, Fan Y, Tong T, Han X, Du Y. Non-radical-dominated catalytic degradation of bisphenol A by ZIF-67 derived nitrogen-doped carbon nanotubes frameworks in the presence of peroxymonosulfate. *Chem. Eng. J.* **336**, 721-731 (2018).
4. Jawad A, et al. Tuning of Persulfate Activation from a Free Radical to a Nonradical Pathway through the Incorporation of Non-Redox Magnesium Oxide. *Environ. Sci. Technol.* **54**, 2476-2488 (2020).
5. Liu T, Zhang D, Yin K, Yang C, Luo S, Crittenden JC. Degradation of thiacloprid via unactivated peroxymonosulfate: The overlooked singlet oxygen oxidation. *Chem. Eng. J.* **388**, 124264 (2020).
6. Lu T, Chen F. Multiwfn: a multifunctional wavefunction analyzer. *J. Comput. Chem.* **33**, 580-592 (2012).
7. Liu T, et al. Unexpected Role of Nitrite in Promoting Transformation of Sulfonamide Antibiotics by Peracetic Acid: Reactive Nitrogen Species Contribution and Harmful Disinfection Byproduct Formation Potential. *Environ. Sci. Technol.* **56**, 1300-1309 (2022).
8. Chu C, et al. Cooperative pollutant adsorption and persulfate-driven oxidation on hierarchically ordered porous carbon. *Environ. Sci. Technol.* **53**, 10352-10360 (2019).
9. Zhang S, et al. Membrane-Confined Iron Oxochloride Nanocatalysts for Highly Efficient Heterogeneous Fenton Water Treatment. *Environ. Sci. Technol.* **55**, 9266-9275 (2021).
10. Rothenberger G, Grätzel M. Effects of spatial confinement on the rate of bimolecular reactions in organized liquid media. *Chem. Phys. Lett.* **154**, 165-171 (1989).
11. Yang Z, Qian J, Yu A, Pan B. Singlet oxygen mediated iron-based Fenton-like catalysis under nanoconfinement. *Proc. Natl. Acad. Sci.* **116**, 6659-6664 (2019).
12. Yang Z, Qian J, Shan C, Li H, Yin Y, Pan B. Toward selective oxidation of contaminants in aqueous systems. *Environ. Sci. Technol.* **55**, 14494-14514 (2021).
13. Chen Y, Miller CJ, Waite TD. Heterogeneous Fenton Chemistry Revisited: Mechanistic

- Insights from Ferrihydrite-Mediated Oxidation of Formate and Oxalate. *Environ. Sci. Technol.* **55**, 14414–14425 (2021).
14. Freer PC. On the formation, decomposition and germicidal action of benzoyl acetyl and diacetyl peroxides. *Am. Chem. J.* **27**, 161-192 (1902).
 15. Kim J, Huang C-H. Reactivity of peracetic acid with organic compounds: A critical review. *ACS ES&T Water* **1**, 15-33 (2020).
 16. Kampf G, Kampf G. Peracetic acid. *Antiseptic Stewardship. Springer, Cham.* 63-98 (2018).
 17. Marchand PA, Derridj S, Ingrid A. COMMISSION IMPLEMENTING REGULATION (EU) No 916/2014. (2015).
 18. Sharma N, Bhardwaj NK, Singh RBP. Environmental issues of pulp bleaching and prospects of peracetic acid pulp bleaching: A review. *J. Clean. Prod.* **256**, 120338 (2020).
 19. Tchobanoglous G, Burton F, Stensel HD. Wastewater engineering: treatment and reuse. *J. Am. Water. Works. Ass.* **95**, 201 (2003).
 20. Meyer E. Disinfection of sewage waters from rendering plants with peracetic acid. *J. Hyg. Epidemiol. Microbiol. Immunol.* **21**, 266-273 (1976).
 21. U.S. Environmental Protection Agency. Emerging technologies for wastewater treatment and in-plant wet weather management. 2013.
 22. United States Environmental Protection Agency. Combined Sewer Overflow Technology Fact Sheet Chlorine Disinfection. (1999).
 23. Bonetta S, et al. Peracetic Acid (PAA) disinfection: Inactivation of microbial indicators and pathogenic bacteria in a municipal wastewater plant. *Water* **9**, 427 (2017).
 24. Kim J, Wang J, Ashley DC, Sharma VK, Huang C-H. Enhanced Degradation of Micropollutants in a Peracetic Acid–Fe(III) System with Picolinic Acid. *Environ. Sci. Technol.* **56**, 4437-4446 (2022).
 25. Chen F, Liu LL, Wu JH, Rui XH, Chen JJ, Yu Y. Single-Atom Iron Anchored Tubular g-C₃N₄ Catalysts for Ultrafast Fenton-Like Reaction: Roles of High-Valency Iron-Oxo Species and Organic Radicals. *Adv. Mater.* **34**, 2202891 (2022).
 26. Lin W, et al. Pre-exposure of peracetic acid enhances its subsequent combination with ultraviolet for the inactivation of fungal spores: Efficiency, mechanisms, and implications. *Water Res.* **229**, 119404 (2023).

27. Zhang T, Wang T, Mejia-Tickner B, Kissel J, Xie X, Huang C-H. Inactivation of Bacteria by Peracetic Acid Combined with Ultraviolet Irradiation: Mechanism and Optimization. *Environ. Sci. Technol.* **54**, 9652-9661 (2020).
28. Wang Z, et al. Application of cobalt/peracetic acid to degrade sulfamethoxazole at neutral condition: Efficiency and mechanisms. *Environ. Sci. Technol.* **54**, 464-475 (2019).
29. Chen Y, Zhang G, Liu H, Qu J. Confining Free Radicals in Close Vicinity to Contaminants Enables Ultrafast Fenton-like Processes in the Interspacing of MoS₂ Membranes. *Angew. Chem. Int. Ed.* **58**, 8134-8138 (2019).
30. Zhang L, et al. Co–Mn spinel oxides trigger peracetic acid activation for ultrafast degradation of sulfonamide antibiotics: Unveiling critical role of Mn species in boosting Co activity. *Water Res.* **229**, 119462 (2023).
31. Oh W-D, Lua S-K, Dong Z, Lim T-T. High surface area DPA-hematite for efficient detoxification of bisphenol A via peroxymonosulfate activation. *J. Mater. Chem. A.* **2**, 15836-15845 (2014).
32. Oh W-D, Dong Z, Hu Z-T, Lim T-T. A novel quasi-cubic CuFe₂O₄–Fe₂O₃ catalyst prepared at low temperature for enhanced oxidation of bisphenol A via peroxymonosulfate activation. *J. Mater. Chem. A.* **3**, 22208-22217 (2015).
33. Huang G-X, Wang C-Y, Yang C-W, Guo P-C, Yu H-Q. Degradation of Bisphenol A by Peroxymonosulfate Catalytically Activated with Mn_{1.8}Fe_{1.2}O₄ Nanospheres: Synergism between Mn and Fe. *Environ. Sci. Technol.* **51**, 12611-12618 (2017).
34. Li X, Wang Z, Zhang B, Rykov AI, Ahmed MA, Wang J. Fe_xCo_{3-x}O₄ nanocages derived from nanoscale metal–organic frameworks for removal of bisphenol A by activation of peroxymonosulfate. *Appl. Catal. B* **181**, 788-799 (2016).
35. Li X, et al. Single cobalt atoms anchored on porous N-doped graphene with dual reaction sites for efficient Fenton-like catalysis. *J. Am. Chem. Soc.* **140**, 12469-12475 (2018).
36. Zhu Y, et al. Heterogeneous photo-Fenton degradation of bisphenol A over Ag/AgCl/ferrihydrate catalysts under visible light. *Chem. Eng. J.* **346**, 567-577 (2018).
37. Lyu L, Yan D, Yu G, Cao W, Hu C. Efficient Destruction of Pollutants in Water by a Dual-Reaction-Center Fenton-like Process over Carbon Nitride Compounds-Complexed Cu(II)-CuAlO₂. *Environ. Sci. Technol.* **52**, 4294-4304 (2018).
38. Wu L, et al. Preferential Growth of the Cobalt (200) Facet in Co@N–C for Enhanced Performance in a Fenton-like Reaction. *ACS Catal.* **11**, 5532-5543 (2021).

39. Gao Y, et al. Activity Trends and Mechanisms in Peroxymonosulfate-Assisted Catalytic Production of Singlet Oxygen over Atomic Metal-N-C Catalysts. *Angew. Chem. Int. Ed.* **60**, 22513-22521 (2021).
40. Wang W, et al. The Confined Interlayer Growth of Ultrathin Two-Dimensional Fe₃O₄ Nanosheets with Enriched Oxygen Vacancies for Peroxymonosulfate Activation. *ACS Catal.* **11**, 11256-11265 (2021).
41. Li X, Ao Z, Liu J, Sun H, Rykov AI, Wang J. Topotactic Transformation of Metal–Organic Frameworks to Graphene-Encapsulated Transition-Metal Nitrides as Efficient Fenton-like Catalysts. *ACS Nano* **10**, 11532-11540 (2016).
42. Zhang M, et al. Efficient removal of organic pollutants by metal–organic framework derived Co/C yolk–shell nanoreactors: Size-exclusion and confinement effect. *Environ. Sci. Technol.* **54**, 10289-10300 (2020).
43. Anipsitakis GP, Dionysiou DD. Radical Generation by the Interaction of Transition Metals with Common Oxidants. *Environ. Sci. Technol.* **38**, 3705-3712 (2004).
44. Wang Z, et al. Application of cobalt/peracetic acid to degrade sulfamethoxazole at neutral condition: Efficiency and mechanisms. *Environ. Sci. Technol.* **54**, 464-475 (2019).
45. Hu P, Long M. Cobalt-catalyzed sulfate radical-based advanced oxidation: A review on heterogeneous catalysts and applications. *Appl. Catal. B* **181**, 103-117 (2016).

Reviewer comments, second round

Reviewer #1 (Remarks to the Author):

The authors have addressed all the required revision. I would like to suggest the publication of the manuscript.

Reviewer #2 (Remarks to the Author):

The MS has been revised by the authors and my concerns have been solved and it is now publishable at Nature Communications

Reviewer #3 (Remarks to the Author):

My previous comments have been carefully addressed and no further comments can be provided now.

REVIEWERS' COMMENTS

Reviewer #1 (Remarks to the Author):

The authors have addressed all the required revision. I would like to suggest the publication of the manuscript.

Response: Many thanks to the Reviewer for the recognition of this work.

Reviewer #2 (Remarks to the Author):

The MS has been revised by the authors and my concerns have been solved and it is now publishable at Nature Communications

Response: We thank the Reviewer very much for the thorough review, which help to improve the clarity and quality of our manuscript.

Reviewer #3 (Remarks to the Author):

My previous comments have been carefully addressed and no further comments can be provided now.

Response: We thank the Reviewer very much for the time and effort to improve the quality of our manuscript as well as the encouraging comments.